# Underappreciated contributions of biogenic volatile organic compounds from urban greening to ozone pollution: a high-resolution modeling study

Haofan Wang[1,2,3,4], Yuejin Li[1,2,3], Yiming Liu[1,2,3]*, Xiao Lu[1,2,3], Yang Zhang[5], Qi Fan[1,2,3]*, Chong Shen[6], Senchao Lai[7], Yan Zhou[7], Tao Zhang[7], Dingli Yue[7]

[1]*School of Atmospheric Sciences, Sun Yat-sen University, and Southern Marine Science and Engineering Guangdong Laboratory (Zhuhai), Zhuhai, 519082, P. R. China*

[2]*Guangdong Provincial Field Observation and Research Station for Climate Environment and Air Quality Change in the Pearl River Estuary, Guangzhou, 510275, P. R. China*

[3]*Guangdong Province Key Laboratory for Climate Change and Natural Disaster Studies, Sun Yat-sen University, P. R. China*

[4]*Centre for Atmospheric Science, Department of Earth and Environmental Sciences, The University of Manchester, Manchester, United Kingdom*

[5]*College of Resources and Environment, Chengdu University of Information Technology, Chengdu, Sichuan, China*

[6]*Guangzhou Climate and Agrometeorology Center, Guangzhou 511430, China*

[7]*National Key Laboratory for Regional Air Quality Monitoring of Environmental Protection/Guangdong Ecological Environment Monitoring Center, Guangzhou 510308, P. R. China*

*Correspondence to*: Yiming Liu (liuym88@mail.sysu.edu.cn), Qi Fan (eesfq@mail.sysu.edu.cn)

# Abstract

Urban Green Spaces (UGS), such as parks, and gardens, are widely promoted as a strategy for improving the urban atmosphere and environmental health. However, this study reveals that it can exacerbate urban ozone ($O_3$) levels under certain conditions, as demonstrated by a September 2017 study in Guangzhou, China. Utilizing the Weather Research and Forecasting Model with the Model of Emissions of Gases and Aerosols from Nature (WRF-MEGAN) and the Community Multiscale Air Quality (CMAQ) model with a high horizontal resolution (1 km), we assessed the impact of UGS-related biogenic volatile organic compound (BVOC) emissions on urban $O_3$. Our findings indicate that the UGS-BVOC emissions in Guangzhou amounted to 666 Gg (~90 Mg/km$^2$), with isoprene (ISOP) and monoterpene (TERP) contributing remarkably to the total UGS-BVOC emissions. In comparison to anthropogenic VOC (AVOC) and BVOC emissions, UGS-BVOC emissions account for approximately 33.45% in the city center region. Incorporating UGS-BVOC emissions into the model significantly reduces the underestimation of ISOP levels compared to observations. The study shows improvements in simulation biases for $NO_2$, from 3.27 to 2.81 ppb, and for $O_3$, from 3.62 to -0.75 ppb. UGS-BVOC and UGS-LUCC (land use cover changes) integration in the air quality model notably enhances surface monthly mean $O_3$ predictions by 1.7-3.7 ppb (+3.8-8.5%) and contributes up to 8.9 ppb (+10.0%) to MDA8 $O_3$ during $O_3$ pollution episodes. Additionally, UGS-BVOC emissions alone increase the monthly mean $O_3$ levels by 1.0-1.4 ppb (+2.3-3.2%) in urban areas and contribute up to 2.9 ppb (+3.3%) to MDA8 $O_3$ levels during $O_3$ pollution episodes. These impacts can extend to surrounding suburban and rural areas through regional transport, highlighting the importance of accurately accounting for UGS-BVOC emissions to better understand and manage their impact on regional air quality.

# Keywords

Urban green space; BVOC; Ozone; Land use cover; CMAQ; MEGAN

# 1. Introduction

Exposure to air pollution now accounts for more fatalities than malaria, tuberculosis, and HIV/AIDS combined (Lelieveld et al., 2020). As a result, the World Health Organization has declared air pollution the most

significant environmental threat to human health (WHO, 2021). Notably, over 70% global health burden of air pollution stems from human-made emissions, leading to a policy focus predominantly on reducing these emissions (Chowdhury et al., 2022; Lelieveld et al., 2019). Despite proactive measures to curb anthropogenic emissions, the incidence of ozone episodes is escalating alongside rapid urbanization (Lu et al., 2020; Yim et al., 2019). Numerous studies have investigated the effects of land use cover changes (LUCC) on air quality during urbanization using numerical models and the majority of these studies conclude that urbanization exacerbates air pollution (Qiu et al., 2023; Wang et al., 2022). However, such studies that depend on numerical models usually face the coarse-resolution land use cover data limitation (Ma et al., 2022, 2019), which leads these studies to frequently overlook a passive abatement approach distinct from reducing anthropogenic sources—namely, the cultivation of urban green spaces (UGS) (Cohen et al., 2017).

The widely accepted notion that UGS can enhance air quality is substantiated by various strands of literature, including public health (Burnett et al., 2018), urban planning (Solomon, 2007), and ecosystem services (Lohmann et al., 2010). This concept is not only prevalent in scholarly circles but also gains traction in popular media and is echoed in international standards and policy frameworks. For instance, the United Nations System of Environmental-Economic Accounting advocates for vegetation as a nature-based approach to mitigate air pollution (Le Page et al., 2015). Vegetation primarily contributes to air pollution reduction through two mechanisms: deposition and dispersion (Shindell et al., 2012). Deposition involves the absorption of air pollutants onto vegetative surfaces, while dispersion refers to the reduction of air pollutant concentrations through aerodynamic effects caused by vegetation (Tiwari and Kumar, 2020; N. Wang et al., 2019a). Notably, Ramanathan et al. (2001) reported that dispersion effects are significantly more impactful than deposition, exceeding it by an order of magnitude via a radiative forcing modeling method.

However, the efficacy of dispersion effects resulting from UGS-LUCC in reducing air pollution is not straightforward. These effects can, under certain conditions, even increase local air pollution concentrations. These conditions are influenced by several factors, such as the specific structure of the UGS vegetation properties (e.g., height, leaf density), the site context (e.g., street canyon geometry, proximity to emission sources), and prevailing meteorological conditions (e.g., wind speed and direction) (Jin et al., 2017; Tomson et al., 2021; Yang et al., 2020). For example, dense tree canopies might impede ventilation in urban street canyons, while porous vegetation barriers in open-road settings could potentially intensify roadside air pollution concentrations (Chen et al., 2021; Jin et al., 2014). Furthermore, Seinfeld et al., (1998) underscores the complexity of these interactions, and demonstrated that vegetation could exert nonlinear effects on

meteorological processes. These effects are particularly evident in their impact on the Planetary Boundary Layer Height (PBLH) and the turbulent transport and advection of pollutants, which in turn influence dispersion conditions.

UGS also have a complex role in air quality due to their production of biogenic volatile organic compounds (BVOCs). For instance, in cities like Los Angeles, the UGS-BVOC emissions contribute to a quarter of the secondary organic aerosol formation on hot days (Schlaerth et al., 2023). While Guenther et al., (2012) noted that the majority of BVOC emissions are from natural land cover, Ma et al., (2022) indicates that in metropolitan areas, the UGS-BVOC emissions can be significantly higher, ranging from 1 to 30 times those from natural land use cover. This evidence suggests a dual nature of UGS vegetation in urban environments: it can mitigate air pollution under certain conditions, but conversely, there is substantial experimental and modeling evidence showing it can exacerbate pollution under different circumstances (Allen and Ingram, 2002; Burnett et al., 2018; Cohen et al., 2017). Moreover, metropolitan areas often encounter VOC-limited conditions, or $NO_x$-saturation, where even minimal BVOC emissions can lead to notable $O_3$ production (P. Wang et al., 2019). Additionally, urban areas typically experience higher temperatures than their surrounding natural landscapes due to the urban heat island effect (Masson-Delmotte et al., 2021). This increase in temperature is likely to further amplify the UGS-BVOC emissions (Zhou et al., 2015), influencing $O_3$ concentrations significantly. This interaction might explain why many regional numerical models underestimate urban surface ozone levels, as they often lack high-resolution land use cover data necessary to accurately estimate the UGS-BVOC emissions (Qiu et al., 2023; Wang et al., 2021; Wu et al., 2020).

Currently, there is a growing research interest in characterizing the air quality impacts of UGS. While Arghavani et al., (2019) investigated the effects of UGS on gaseous air pollutants in Tehran using the WRF-Chem model, their focus was on the impact of meteorological changes on $O_3$ resulting from UGS (i.e., UGS-LUCC effects), rather than the UGS-BVOC emissions effects on $O_3$. In contrast, Schlaerth et al., (2023a) addressed the influence of the UGS-BVOC emissions on $O_3$ in Los Angeles and their findings indicate that the UGS-BVOC emissions may increase $O_3$ by 0.95 ppb during the daytime and decrease it by 0.41 ppb at night. Despite Schlaerth et al., (2023a) illustrating the significance of the UGS-BVOC emissions on $O_3$ concentrations, they did not investigate the impact of the UGS-LUCC effects.

Surface $O_3$ is generally formed through chemical reactions of VOCs and $NO_x$ in the presence of sunlight. The nonlinear correlation between $O_3$ and concentrations of BVOC and $NO_x$ underscores the importance of

examining potential interactions between the UGS-BVOC emissions and anthropogenic emissions. Furthermore, recent studies have highlighted the significance of the UGS-LUCC effects and the UGS-BVOC emissions effects. Given the rise in urban $O_3$ pollution, investigating the influence of the UGS-LUCC effects and the UGS-BVOC emissions effects on $O_3$ can assist in rationalizing UGS planning and formulating air quality mitigation strategies. However, there is a lack of quantification regarding the combined effects of UGS-LUCC and UGS-BVOC emissions on $O_3$.

Situated in South China, Guangzhou (Figure 1) is one of the rapidly expanding cities in China since the initiation of the reform and opening-up policy, undergoing swift urbanization (Yao and Huang, 2023). Being a key city in the Guangdong-Hongkong-Macao Greater Bay Area, Guangzhou places significant emphasis on UGS development. In this study, we aim to reconstruct the leaf area index (LAI) dataset for urban areas and estimate the UGS-BVOC emissions utilizing the Model of Emissions of Gases and Aerosols from Nature version 3.1 (MEGANv3.1) (Guenther et al., 2020a). Subsequently, employing the Weather Research and Forecast model version 4.1.1 (WRFv4.1.1) (Salamanca et al., 2011) – Community Multiscale Air Quality model version 5.4 (CMAQv5.4) (https://zenodo.org/record/7218076, last accessed: June 3, 2023), we intend to estimate the improvements of the CMAQ simulation performance from considering UGS-LUCC and UGS-BVOC and investigate the UGS-LUCC effects, the UGS-BVOC emissions effects, and their combined impacts on $O_3$ over Guangzhou by configuring sensitivity cases.

## 2. Methods and data

### 2.1 Leaf area index and land cover dataset

The default LAI dataset to drive the MEGANv2.1 model which can used for MEGANv3.1 is derived from the enhanced Moderate Resolution Imaging Spectroradiometer (MODIS) /MOD15A2H in 2003 with 1 km spatial resolution (Myneni et al., 2015). As MODIS/MOD15A2H assigns an LAI value of 0 to urban areas, MEGANv3.1 compensates by averaging the LAI values in the vicinity of the urban area. However, this approach introduces considerable uncertainty in the estimation of UGS-BVOC emissions. Hence, we opted for the Global Land Surface Satellite (GLASS) LAI product for MEGANv3.1 in 2017 with 500-m spatial resolutions, derived from MODIS surface reflectance data using the bidirectional long short-term memory (Bi-LSTM) model, which leverages existing global LAI products (Ma and Liang, 2022) and effectively incorporates the temporal and spectral information of MODIS surface reflectance. Consequently, the valid

values of this data extend to urban areas, making it suitable for simulating the UGS-BVOC emissions.

In this study, UGS are delineated as vegetation areas within the urban grid, and the urban grids are derived
from MODIS/MCD12Q1 (Friedl and Sulla-Menashe, 2019) in 2017, which corresponds to the simulation
period with 500 m spatial resolution. Furthermore, a high-resolution (10 m) land cover dataset in 2017 was
also obtained from the Geographic Remote Sensing Ecological Network Platform (accessible at
http://www.gisrs.cn/infofordata?id=1c089287-909e-4394-b07f-c7004be60884, last accessed: 20/11/2023)
and was employed to depict the spatial patterns of UGS. The processed land cover dataset is illustrated in
Figure S1. Meanwhile, the use of high-resolution land use cover data is pivotal for accurately depicting the
intricate details of land use cover, especially in areas broadly classified as urban by coarse-resolution data (i.e.,
MCD12Q1) and this refined approach allows for a more precise differentiation of UGS. Specifically, we
maintain a consistent urban area definition across both land use cover datasets, anchored by the urban
delineation provided by the MCD12Q1 dataset. However, the coarse resolution of MCD12Q1 is insufficient
for detailed spatial characterization of UGS. To address this limitation, we employ the high-resolution dataset
to refine the characterization of non-urban surfaces within the urban boundaries (i.e., UGS) defined by
MCD12Q1. This approach yields a sophisticated land cover dataset with 10 m spatial resolution that retains
the urban extent delineated by MCD12Q1 while incorporating detailed representations of UGS absent in the
original dataset. Consequently, while both datasets encompass identical urban extents, the default dataset lacks
representations of UGS, in contrast to the high-resolution dataset, which includes detailed depictions of UGS.
**2.2 MEGANv3.1 configuration**
The calculation of BVOC emissions was performed utilizing MEGANv3.1 (accessible at
https://bai.ess.uci.edu/megan, last accessed: 21 November 2023), which is a newly updated version.
MEGANv3.1 estimates BVOC emissions as the product of an emission factor and an emission activity factor
( Guenther et al., 2020a):
$$E = EF \times \gamma \quad\quad\quad\quad\quad\quad\text{(Eq. 1)}$$


In this equation, E is the net emission flux ($\mu g\ m^{-2}\ h^{-1}$), and EF is the weighted average of the emission
factor ($\mu g\ m^{-2}\ h^{-1}$) for each vegetation type calculated by Emission Factor Processor (EFP). The emission
activity factor ($\gamma$) considers emission responses to changes in environmental and phenological conditions.
Compare with earlier versions, $\gamma$ in MEGANv3.1 adds quantifications for responses to high and low
temperature, high wind speed, and air pollution ($O_3$).

$$\gamma = LAI \times \gamma_{TP} \times \gamma_{LA} \times \gamma_{SM} \times \gamma_{HT} \times \gamma_{LT} \times \gamma_{HW} \times \gamma_{CO_2} \times \gamma_{BD} \times \gamma_{O_3} \qquad \text{(Eq. 2)}$$


In this equation, the activity factor denotes the emission response to canopy temperature/light($\gamma_{TP}$), leaf age
($\gamma_{LA}$), soil moisture ($\gamma_{SM}$), high temperature ($\gamma_{HT}$), low temperature ($\gamma_{LT}$), high wind speed ($\gamma_{HW}$), ambient
$CO_2$ concentration ($\gamma_{CO_2}$), bidirectional exchange ($\gamma_{BD}$), $O_3$ exposure ($\gamma_{O_3}$), and Leaf Area Index (LAI). In
this study, $\gamma_{CO_2}$ was not considered in the BVOC emission estimation. The MEGANv3.1 approach can
calculate the emissions at each canopy level as the product of the emission factor and emission activity at
each level.

Hence, the input data to drive MEGANv3.1 comprises meteorological variables (e.g., temperature, solar
radiation, relative humidity, soil moisture), LAI, and three types of land cover data (i.e., ecotype, growth
form, and relative vegetation composition for each ecotype/growth form). Meanwhile, the growth form
datasets in MEGANv3.1 contain considerations of evergreen broadleaf forests, grasslands, and crops, which
cover all types of UGS in Guangzhou city (Figure S1). Meteorological data are obtained from the WRF
simulation results, and the LAI dataset is detailed in Section 2.1 as well as additional default land cover data
provided by MEGANv3.1 were employed.
**2.3 WRF-CMAQ and Case Configuration**
Both the WRFv4.1.1 model and the CMAQv5.4 model are compiled and operated on a server with a Linux
environment. The WRFv4.1.1 model was employed to simulate meteorological conditions, utilizing initial and
boundary conditions sourced from the NCEP 1° × 1° Final (FNL) reanalysis dataset (National Centers for
Environmental Prediction, National Weather Service, NOAA, U.S. Department of Commerce, 2000). As
illustrated in Figure S2, four nested domains with horizontal resolutions of 27, 9, 3, and 1 km, respectively,
were employed. The outermost domain encompasses mainland China, while the innermost domain zooms in
Guangzhou city, and the physical parameterization configured for the WRF simulation is listed in Table S1.
CMAQv5.4 utilized meteorological fields provided by WRF to model $O_3$ concentrations. The initial and
boundary conditions for the CMAQ model were derived from the default profiles representing a clean
atmosphere. In addition, we acquired anthropogenic emissions for the CMAQ domain from the Multi-
resolution Emission Inventory for China (MEIC) 2017 developed by Tsinghua University, which contains

monthly gridded (0.25° × 0.25°) emissions information for anthropogenic emissions. Moreover, the CMAQ model was configured with the Carbon Bond chemical mechanism (CB06) (Luecken et al., 2019) and AERO7 (Pye et al., 2017). In this study, we incorporated the Modular Emission Inventory Allocation Tools for the Community Multiscale Air Quality model (MEIAT-CMAQ, https://github.com/Airwhf/MEIAT-CMAQ, last accessed: February 27, 2024) to allocate spatial and species-specific emissions within the raw inventories, addressing discrepancies in resolution and species compared to the modeled configurations. Moreover, MEIAT-CMAQ can directly generate the hourly model-ready emission files for CMAQ via temporal allocation. The model simulation spanned a month, from 21 August 2017 to 30 September 2017. To mitigate bias resulting from meteorological and chemical drift, the initial 10 days of this simulation were designated as spin-up and were not included in the analysis for this study. Given the spatial heterogeneity in the distribution of UGS across different areas, this study categorizes Guangzhou into city center, suburban, and rural regions (Figure 1). Specifically, the city center areas comprise Haizhu (HZ), Liwan (LW), Yuexiu (YX), and Tianhe (TH) districts. The city center region has more UGS areas due to the higher urban land use and land cover fraction (Figure S1) compared to the suburban and rural regions. The suburban areas encompass Huangpu (HP), Baiyun (BY), Panyu (PY), and Nansha (NS) districts. Lastly, the rural regions include Zengcheng (ZC), Conghua (CH), and Huadu (HD) districts. To facilitate clear differentiation between the two sites in the HP region, they have been designated as HP_L and HP_H, respectively.

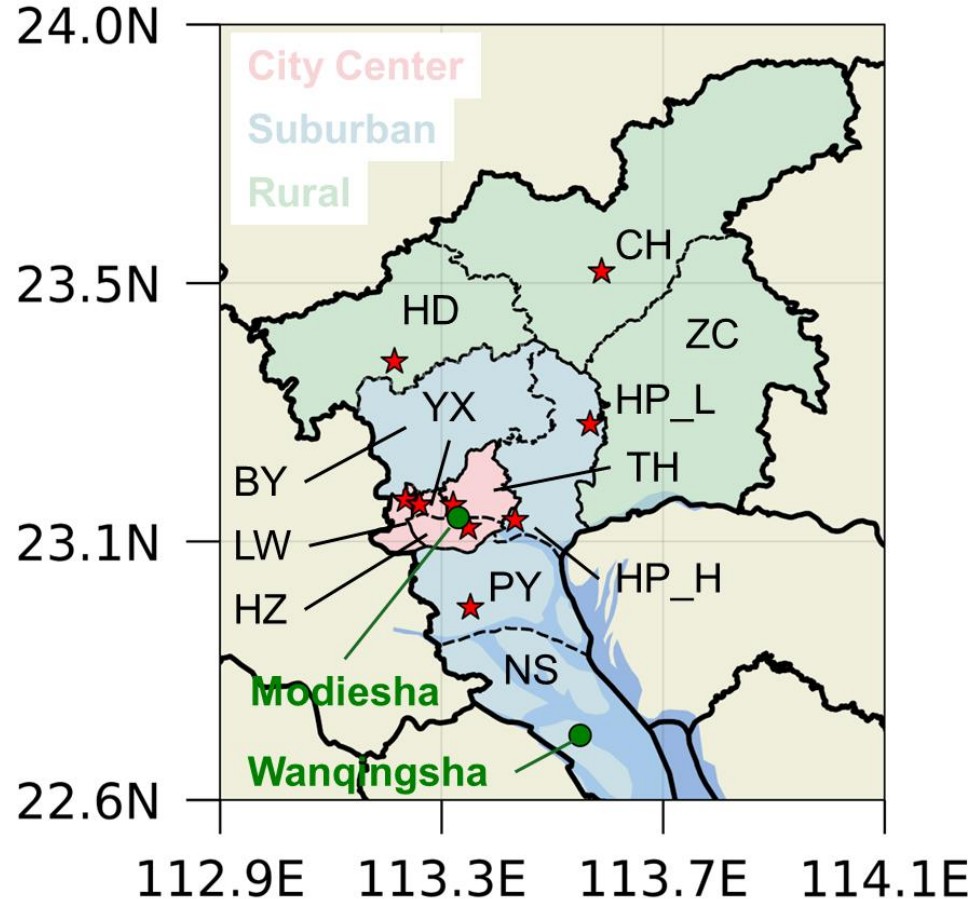

**Figure 1 The innermost domain of WRF-CMAQ with various areas and the air quality station locations map. Modiesha and Wanqingsha**
**are the observation sites for isoprene.**

In this study, four distinct cases, as listed in Table 1 were established to investigate the impacts of UGS-LUCC,
UGS-BVOC, and their combined effects on the ozone simulation. These cases also focused on the performance
of the CMAQ simulation and the influence on $O_3$ episodes. The Gdef_N case considered as the base case,
employs default land use cover data—specifically, data excluding UGS, and uses the LAI dataset with urban
areas omitted (N-LAI). In contrast, the Gdef_Y case is similar to Gdef_N but incorporates the LAI dataset that
includes urban areas (T-LAI). This adjustment allows for the assessment of the UGS-BVOC emission effects
on $O_3$ concentrations. The Ghr_N case mirrors Gdef_N but differs by integrating high-resolution land use
cover data, which encompasses UGS land use cover. This case aims to examine the UGS-LUCC effects on $O_3$
concentrations. Finally, the Ghr_Y case combines high-resolution land use cover data with the LAI dataset
inclusive of urban areas, thereby enabling an exploration of the combined effects of UGS-BVOC emissions
and UGS-LUCC on $O_3$ concentrations.



**Table 1 Case configurations**

| Name | LC dataset | LAI dataset | Description |
|---|---|---|---|
| Gdef_N | Default data | N-LAI | Base |
| Gdef_Y | Default data | T-LAI | UGS-BVOC effects |
| Ghr_N | High-resolution data | N-LAI | UGS-LUCC effects |
| Ghr_Y | High-resolution data | T-LAI | combined effects |

**2.4 Observation Dataset**
We use the hourly ground-level meteorological observations, encompassing 2-m temperature (T2) and 10-m
wind speed (WS10), sourced from national basic meteorological stations provided by the Guangdong
Provincial Meteorological Service (Figure S3). Hourly ambient concentrations of $O_3$, CO, and $NO_2$ from
national monitoring stations are gathered from the China National Environmental Monitoring Centre
(CNEMC; http://www.cnemc.cn, last assess: 24 December 4, 2024). The real-time hourly concentration of
$O_3$ was measured by the ultraviolet absorption spectrometry method and differential optical absorption
spectroscopy at each monitoring site. $NO_2$ concentrations are measured by the molybdenum converter method
known to have positive interferences from $NO_2$ oxidation products (Dunlea et al., 2007).The instrumental
operation, maintenance, data assurance, and quality control were properly conducted based on the most recent
revisions of China Environmental Protection Standards (Zhang and Cao, 2015), and the locations of these air
quality stations are depicted in Figure 1. Additionally, meteorological data also undergo thorough quality
control. Subsequently, they are utilized to assess the model performance of WRF-CMAQ.

For the isoprene (ISOP) evaluation, we use observation data from the Modiesha (23.11°N, 113.33°E) and
Wanqingsha (22.71°N, 113.55°E) sites (Figure 1), where an online gas chromatography-mass
spectrometry/flame ionization detector system (GC-FID/MSD, TH 300B, Wuhan) is used to measure VOCs
in the ambient atmosphere. The system has a sampling rate of 60 mL/min for 5 minutes per sample, with a
sampling frequency of once per hour (Meng et al., 2022). The ISOP observation data undergo rigorous quality
control, which can be used for evaluating simulated ISOP concentrations. It is worth noting that the ISOP
observational data for the Modiesha site covers September 2017, while the Wanqingsha site has data coverage
from September 7 to September 30, 2017.

# 3. Results and discussion

## 3.1 Model Evaluation

Evaluation of the WRF-CMAQ model performance is undertaken through comparison against ground-level observations and the evaluation metrics of meteorological parameters are listed in Table S2, which shows that the meteorological fields were faithfully reproduced in this study and can be used to drive the air quality model.

BVOCs are the major sources of ISOP and monoterpene (TERP), rendering the assessment of their concentrations a pivotal method for indirectly verifying the accuracy of BVOC emission estimates. Table 2 delineated within this study presents the mean concentrations of ISOP derived from various cases juxtaposed with the observed average concentrations. This comparative analysis in the Modiesha site reveals that after the incorporation of the UGS-BVOC emissions, there is an augmentation in the ISOP concentration from 0.29 to 0.35 ppb and from 0.23 to 0.29 ppb under distinct land use cover cases (Gdef and Ghr), relative to an observed concentration of 0.34 ppb. Meanwhile, the evaluation at the Wanqingsha site, where the observed mean ISOP concentration was 0.45 ppb from September 7 to September 30, 2017, shows that the modeled ISOP concentrations increased from 0.29 to 0.31 ppb and from 0.27 to 0.29 ppb under distinct land use cover cases (Gdef and Ghr) when UGS-BVOC emissions were included. This increment signifies a substantial diminution in the discrepancy between the modeled and observed concentrations attributable to the UGS-BVOC emissions. Analogously, the integration of the UGS-BVOC emissions yields a refinement in the estimation accuracy of ISOP concentrations at the Modiesha site, as evidenced by a reduced bias.

These findings reveal that ISOP concentrations are underestimated by 16.4% and 34.7% in the Modiesha and Wanqingsha sites when UGS-BVOCs are excluded, respectively, suggesting the important role of UGS-BVOCs emissions in modeling. Moreover, numerous studies highlight the significant role of ISOP in $O_3$ formation within the Pearl River Delta (PRD) region, including Guangzhou. For instance, Zheng et al., (2009) demonstrated that ISOP has the highest ozone formation potential among all VOCs. Therefore, incorporating UGS-BVOCs into ISOP concentration estimates is crucial for accurately modeling regional $O_3$ levels.


**Table 2 The evaluation results for the monthly mean ISOP concentrations. The "Gdef_N", "Gdef_Y", "Ghr_N", and "Ghr_Y" columns show the various metrics from comparing the hourly observation and simulation values during September 2017 for the Modiesha site and 7 September 2017 to 30 September 2017 for the Wanqingsha site.**

| Site name | Metrics | Gdef_N (ppb) | Gdef_Y (ppb) | Ghr_N (ppb) | Ghr_Y (ppb) |
|---|---|---|---|---|---|
| Modiesha | Sim. | 0.29 | 0.35 | 0.23 | 0.29 |
| | Obs. | 0.34 | 0.34 | 0.34 | 0.34 |
| | MB | -0.06 | 0.01 | -0.11 | -0.05 |
| | NME | 76.0% | 68.7% | 73.6% | 66.2% |
| | NMB | -16.4% | 3.5% | -31.3% | -13.1% |
| | R | 0.44 | 0.46 | 0.37 | 0.39 |
| Wanqingsha | Sim. | 0.29 | 0.31 | 0.27 | 0.29 |
| | Obs. | 0.45 | 0.45 | 0.45 | 0.45 |
| | MB | -0.15 | -0.14 | -0.17 | -0.15 |
| | NME | 58.9% | 56.8% | 60.4% | 58.1% |
| | NMB | -34.7% | -30.6% | -38.7% | -34.8% |
| | R | 0.35 | 0.39 | 0.34 | 0.4 |


Additionally, various statistical metrics were used to assess the performance of hourly $O_3$, MDA8 $O_3$, and $NO_2$
concentrations from the CMAQ simulation (Emery et al. 2017). These metrics comprise the correlation
coefficient (R), normalized mean bias (NMB), and normalized mean error (NME). The formulas for these
metrics are listed in Table S3. As shown in Table 3, the modeling performance for all cases are reasonably,
albeit with some degree of underestimation. Despite these discrepancies, the model demonstrates sufficient
reliability and can be effectively used in the subsequent study. Meanwhile, the MBs of MDA8 $O_3$ across
various cases indicate a substantial improvement in the CMAQ simulation when UGS-BVOC, UGS-LUCC,
and their combined effects are considered. Specifically, the MB values decrease from -2.16 ppb in the Gdef_N
case to -0.26 ppb in the Ghr_Y case, demonstrating that incorporating UGS-BVOC, UGS-LUCC, and their
combined effects can enhance the accuracy of predicted daytime $O_3$ concentrations. In addition, we also
evaluate the simulation performance for $NO_2$ in each case and the results suggest that all models have R above
0.63, and while there is some overestimation, the NMB is 15.0%, 15.2%, 13.0%, and 13.2% for Gdef_N,
Gdef_Y, Ghr_N, and Ghr_Y, respectively. It should be emphasized that integrating UGS-BVOC into the
modeling process can slightly improve the accuracy of $NO_2$ predictions, reducing the MB from 3.27 to 3.24
ppb, and from 2.84 to 2.81 ppb for Gdef and Ghr cases, respectively. The improvement in $NO_2$ predictions is
attributed to the increased involvement of $NO_2$ in $O_3$ formation caused by the UGS-BVOC emissions, which
reduces simulated $NO_2$ concentrations and narrows its bias against the observation.


**Table 3 Evaluation results of the simulated monthly mean hourly O₃, MDA8 O₃, and hourly NO₂ mixing ratios for each case during**
**September 2017.**

| Pollutant | Case name | Sim (ppb) | Obs (ppb) | MB (ppb) | NMB | NME | R |
|---|---|---|---|---|---|---|---|
| Hourly O₃ | Gdef_N | 28.23 | 30.49 | -2.26 | -6.7% | 23.6% | 0.82 |
| | Gdef_Y | 28.67 | 30.49 | -1.82 | -5.3% | 23.6% | 0.82 |
| | Ghr_N | 28.89 | 30.49 | -1.60 | -4.8% | 22.5% | 0.83 |
| | Ghr_Y | 29.33 | 30.49 | -1.15 | -3.4% | 22.4% | 0.83 |
| MDA8 O₃ | Gdef_N | 60.11 | 62.27 | -2.16 | -3.47% | 21.71% | 0.84 |
| | Gdef_Y | 61.04 | 62.27 | -1.23 | -1.97% | 21.40% | 0.84 |
| | Ghr_N | 61.07 | 62.27 | -1.20 | -1.92% | 21.28% | 0.84 |
| | Ghr_Y | 62.00 | 62.27 | -0.26 | -0.42% | 21.23% | 0.84 |
| Hourly NO₂ | Gdef_N | 24.78 | 21.50 | 3.27 | 15.2% | 45.7% | 0.63 |
| | Gdef_Y | 24.74 | 21.50 | 3.24 | 15.0% | 45.5% | 0.63 |
| | Ghr_N | 24.35 | 21.50 | 2.84 | 13.2% | 43.8% | 0.63 |
| | Ghr_Y | 24.32 | 21.50 | 2.81 | 13.0% | 43.6% | 0.63 |

In terms of O₃, the UGS-BVOC, UGS-LUCC, and their combined effects have various performances in different regions (Table 4). These results indicate that the inclusion of UGS-BVOC emissions remarkable influences MDA8 O₃ and hourly O₃ concentrations in the city center region and this effect, primarily observed when comparing the Gdef_Y with Gdef_N and Ghr_Y with Ghr_N cases, is largely due to the VOC-limited areas prevalent in Guangzhou (He et al., 2024). By integrating the UGS-BVOC emissions into the models (comparing Gdef_Y and Gdef_N cases), the MBs of MDA8 O₃ and hourly O₃ in all regions, including a notable improvement in the city center region from -3.62 to -0.75 ppb and -2.86 to -1.18 ppb, respectively, is reduced. Additionally, the UGS-BVOC emissions slightly enhance R values of MDA8 O₃ and hourly O₃ in the city center and suburban regions, indicating a more accurate the daytime trend and the diurnal cycle representation, respectively. The UGS-LUCC effects, as seen when comparing Ghr_N and Gdef_N cases, also greatly improve model biases and the combined effects of both UGS-BVOC and UGS-LUCC (comparing the Ghr_Y and Gdef_N cases) substantially ameliorate model biases in the city center and suburban regions.

**Table 4 Evaluation results of simulated monthly mean hourly O3 and MDA8 O3 mixing ratios in city center, suburban, and rural areas**
**for each case during September 2017.**

| Variable | Regions | MB (ppb) | | | | R | | | |
|---|---|---|---|---|---|---|---|---|---|
| | | Gdef_N | Gdef_Y | Ghr_N | Ghr_Y | Gdef_N | Gdef_Y | Ghr_N | Ghr_Y |
| MDA8 O₃ | City center | −3.627 | −2.241 | −2.110 | −0.747 | 0.805 | 0.810 | 0.810 | 0.813 |
| | Suburban | −4.076 | −3.251 | −3.210 | −2.376 | 0.737 | 0.743 | 0.717 | 0.727 |
| | Rural | −5.109 | −4.757 | −4.866 | −4.528 | 0.665 | 0.655 | 0.695 | 0.690 |
| Hourly O₃ | City center | -2.862 | -2.292 | -2.086 | -1.520 | 0.800 | 0.802 | 0.811 | 0.812 |
| | Suburban | -3.148 | -2.803 | -2.647 | -2.295 | 0.824 | 0.825 | 0.824 | 0.826 |
| | Rural | -1.184 | -1.630 | -1.375 | -1.164 | 0.742 | 0.741 | 0.751 | 0.750 |

**3.2 Estimation of UGS-BVOC emissions under different land use cover**

This study comprehensively summarizes the UGS-BVOC emissions across various species for all regions in Guangzhou City in September. Given that the variances in the UGS-BVOC emissions due to different land use covers are relatively minor, the primary Table 5 presents emissions driven by the default land use cover. For a detailed breakdown of emissions attributable to varied land use covers, refer to Table S4. A review of the data reveals that TERP and ISOP rank as the highest emitting species with proportions are 20.46% and 31.91% in this study, respectively, aligning with the findings of previous studies (Cao et al., 2022; Guenther et al., 2012b). Furthermore, Table 5 reveals that in September, the UGS-BVOC emissions in Guangzhou amounted to 666 Gg (~90 Mg/km$^2$), with ISOP and TERP contributing remarkably to the total UGS-BVOC emissions. In comparison to anthropogenic VOC (AVOC) and BVOC emissions, UGS-BVOC emissions account for approximately 33.45% in the city center region. Regionally, the suburban region registered the highest UGS-BVOC emissions in Guangzhou, peaking at 367 Gg. This is followed closely by the rural and city center regions, recording emissions of 173 Gg and 125 Gg, respectively.

**Table 5 The summarized table of UGS-BVOC emissions in Guangzhou city in September 2017 via default land use cover (units: Gg).**

| Species | Abbreviations | City center (Gg) | Suburban (Gg) | Rural (Gg) | Total (Gg) |
|---|---|---|---|---|---|
| Acetic acid | AACD | 0.86 | 2.44 | 1.18 | 4.48 |
| Acetaldehyde | ALD2 | 3.46 | 11.57 | 5.83 | 20.86 |
| Formaldehyde | FORM | 0.95 | 3.90 | 2.17 | 7.02 |
| Methanol | MEOH | 12.47 | 41.31 | 20.36 | 74.14 |
| Formic acid | FACD | 2.79 | 7.84 | 3.79 | 14.42 |
| Ethane | ETHA | 2.12 | 8.40 | 4.64 | 15.16 |
| Ethanol | ETOH | 3.63 | 12.13 | 6.11 | 21.87 |
| Acetone | ACET | 6.22 | 21.52 | 11.63 | 39.37 |
| Propane | PRPA | 2.08 | 8.21 | 4.54 | 14.83 |
| Ethene | ETH | 3.97 | 15.64 | 8.64 | 28.25 |
| Isoprene | ISOP | 47.30 | 117.32 | 48.06 | 212.68 |
| Monoterpenes | TERP | 24.07 | 74.85 | 37.51 | 136.43 |
| Alpha pinene | APIN | 11.26 | 30.07 | 13.16 | 54.49 |
| Methane | ECH4 | 0.04 | 0.14 | 0.08 | 0.26 |
| Sesquiterpenes | SESQ | 4.31 | 11.97 | 5.95 | 22.23 |
| Total | Total | 125.53 | 367.31 | 173.65 | 666.49 |

Figure 2A provides a detailed illustration of the UGS-BVOC emissions across various regions in Guangzhou City, driven by default land use cover data, and compares these with the estimates derived from high-resolution land use cover data, which presents that the suburban region exhibits the highest UGS-BVOC emissions among the three studied regions, totaling 413.47 Gg. This predominance is linked to the larger extent of UGS in the suburban region, as depicted in Figure 5A, while the emissions in the city center and rural regions are

reported at 137.69 Gg and 198.64 Gg, respectively. Moreover, UGS-LUCC is instrumental in modulating BVOC emissions, leading to an uptick in the city center and rural regions while precipitating a decline in the suburban region. Notably, a slight increase in solar radiation (SOL_RAD) by 0.05% (Figure 2C), attributable to a reduced urban fraction in the Ghr dataset, results in augmented solar exposure. Concurrently, a marginal reduction in surface temperature (SFC_TMP) by 0.02% (Figure 2D), facilitated by increased vegetation albedo cooling effects, underpins the decrease in UGS-BVOC emissions within suburban regions. This phenomenon underscores the critical role of lowered SFC_TMP—driven by vegetation's higher albedo—in curtailing emissions stemming from UGS-LUCC. Moreover, in the city center contexts, the diminished urban fraction enhances SOL_RAD and SFC_TMP, promoting higher emissions, a trend mirrored to a lesser extent in the rural region following the update of land use cover data to Ghr. Figure 2B offers a clear depiction of the proportion of UGS-BVOC emissions relative to non-UGS area BVOC emissions in each region of Guangzhou City, which presents that the UGS-BVOC emissions in the city center region constitute 57.34% of the total BVOC emissions in this region because of the larger urban proportions in the city center region (Figure 5), while the UGS-BVOC emission proportion in suburban and rural are 19.44% and 1.86% respectively. This indicates a significant contribution of the UGS-BVOC emissions in the the city center region. Furthermore, when examining the relative differences in the BVOC emissions resulting from various land use covers across the city, the changes are found to be minimal, which suggests that meteorological alterations from land use cover do not majorly influence the proportion of the UGS-BVOC emissions emanating in Guangzhou. Thus, factors other than land use changes might be more critical in shaping the distribution and intensity of the UGS-BVOC emissions in urban settings.

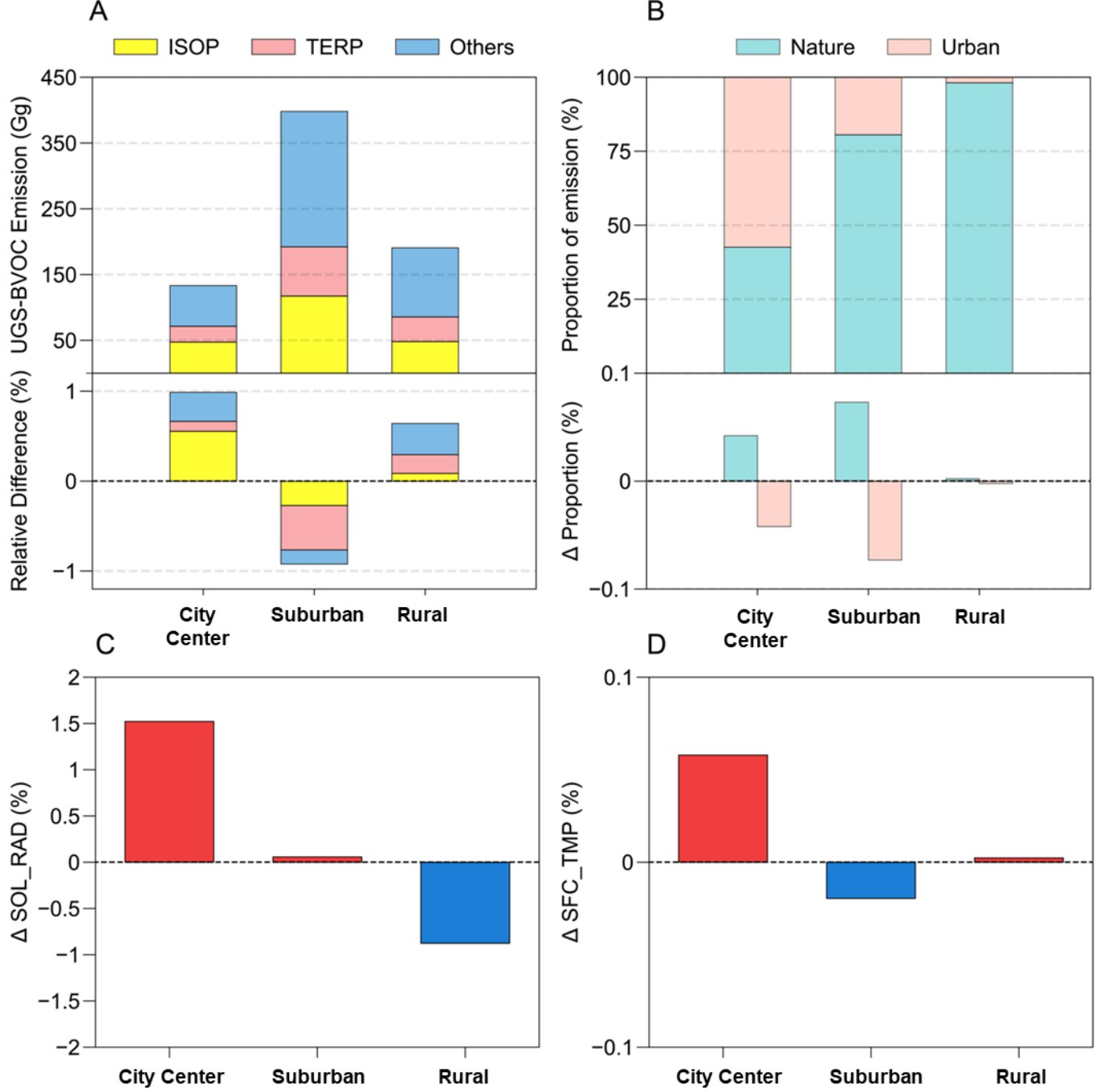

Figure 2 The UGS-BVOC emissions of each species and relative difference (Ghr - Gdef) from various land use cover (a), the proportion of emissions from urban and nature and the relative proportion difference (Ghr - Gdef) from various land use cover (b), the relative difference of solar radiation (C), and surface temperature (D) driven via various land use cover datasets. All values in these figure are during September 2017.

Figure 3A-B collectively highlight the patterns of the UGS-BVOC emissions across different land use covers, pinpointing the emission hotspots in city center and suburban regions, which effectively illustrate how land use cover influences the UGS-BVOC emissions in various parts of the city. Additionally, Figure 3C delves into the disparities in the UGS-BVOC emissions attributed to different land use cover datasets. It reveals that the variations in emissions are predominantly concentrated in the identified hotspots. Moreover, Figure 3 C indicates that employing high-resolution land cover data typically results in marginally higher estimates of the

UGS-BVOC emissions, with an increase ranging between 0.8% to 2.9%. Figure 5E-F illustrate that despite a marginal reduction in solar radiation within the city center region, a corresponding minor temperature elevation modestly boosts UGS-BVOC emissions, which presents that the increase in temperature from UGS-LUCC causes the rise of the UGS-BVOC emissions.

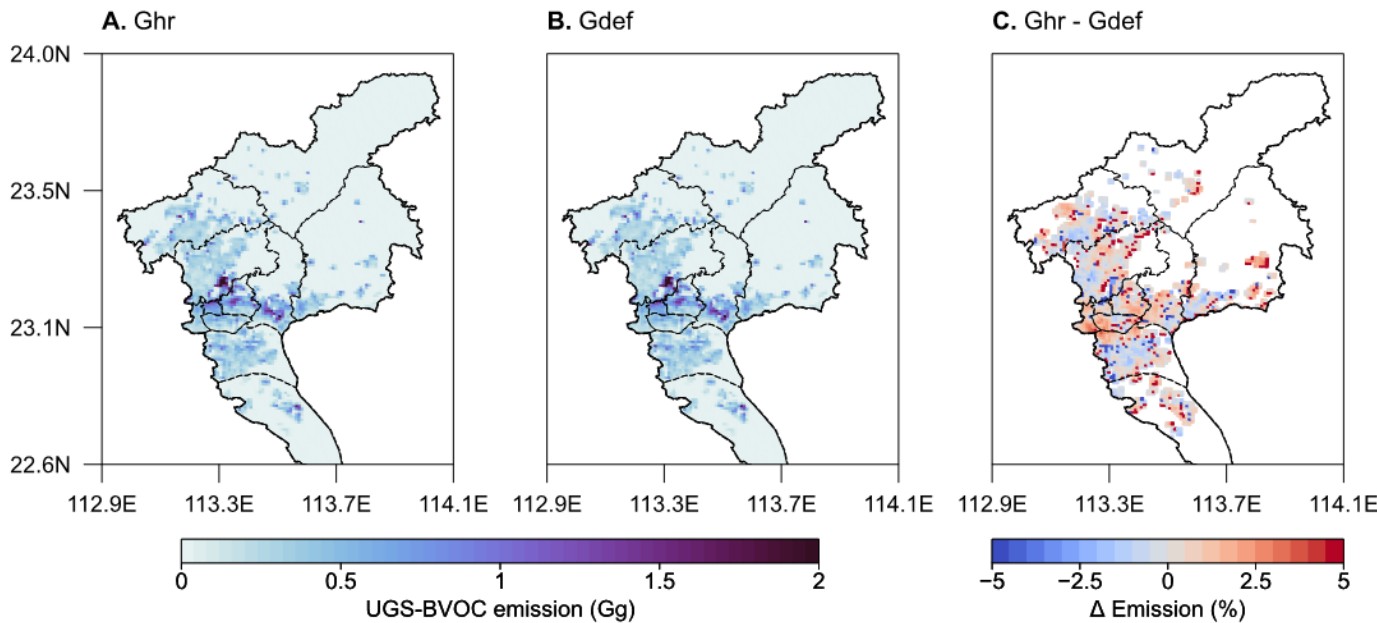

**Figure 3 The UGS-BVOC emission maps in September 2017 from default (A) and high-resolution (B) land use cover, and the differences of various UGS-BVOC emissions (C).**

As illustrated in Figure S1, UGS in Guangzhou comprises three primary types of vegetation: evergreen broadleaf forests, which are composed of Evergreen Broadleaf Trees (EBTs), cropland, and grasslands. This classification has enabled a more nuanced understanding of how different types of UGS vegetation influence UGS-BVOC emissions. Figure 4 reveals that EBTs predominate the urban vegetation landscape in Guangzhou and are associated with higher rates of UGS-BVOC emissions as their coverage increases. Conversely, an increase in the proportion of cropland correlates with reduced UGS-BVOC emissions, highlighting its minimal contribution to the overall UGS-BVOC emissions of Guangzhou. Grasslands exhibit a variable impact on BVOC emissions; when they constitute over 80% of the UGS, the emission rates are relatively low. However, when grassland coverage ranges between 60-80%, its BVOC emissions surpass those from cropland within the same percentage range. Overall, EBTs emerge as the primary contributors to UGS-BVOC emissions, with grasslands and croplands making lesser contributions.

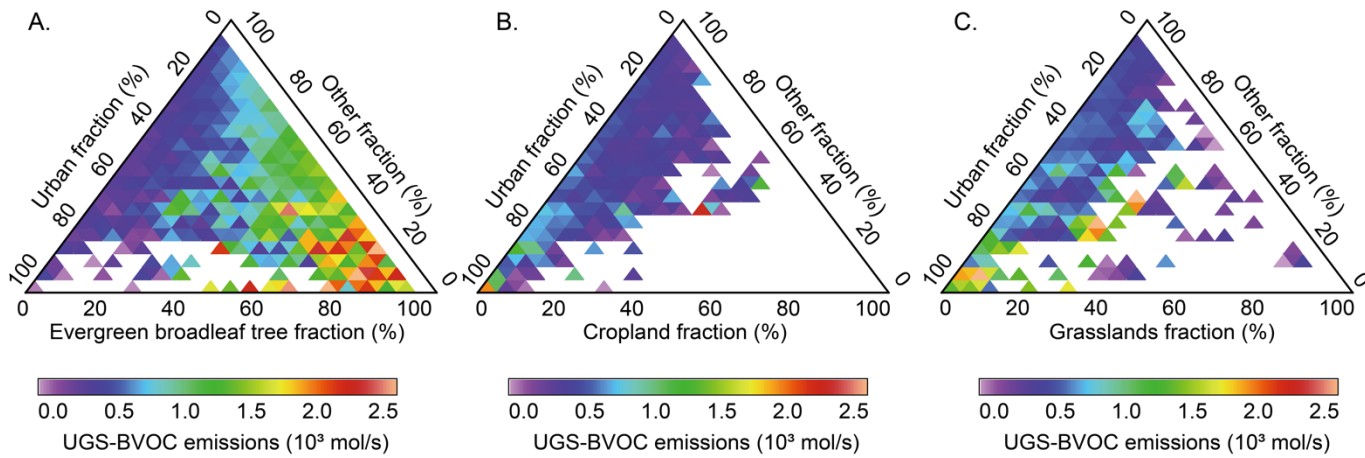

**Figure 4 Ternary heat map for various vegetation in UGS with the UGS-BVOC emission rate and the invalid value in this figure represents no UGS-BVOC emission.**

In addition to the proportion of UGS, the UGS-BVOC emissions in Guangzhou city are significantly influenced by meteorological factors such as surface temperature and solar radiation (Guenther et al., 2020b). To elucidate the spatial heterogeneity of the UGS-BVOC emissions, this study analyzes variations in these key factors. The simulation results depicted in Figure 5A show the distribution pattern of UGS, which are predominantly located in the city center region, which account for a higher percentage of the UGS-BVOC emissions compared to others. Interestingly, as indicated in Figure 5B, the city center region receives less solar radiation than other regions likely due to the shading effect of urban canopies. Conversely, the city center region exhibits elevated temperatures attributable to the urban heat island effect, leading to an increase in UGS-BVOC emissions. Thus, while the distribution of UGS contributes to the variation in the UGS-BVOC emissions across different regions, the more significant factor is the enhanced UGS-BVOC emission due to higher temperatures in densely urbanized areas. The spatial dynamics of the UGS-BVOC emissions are significantly shaped by two key meteorological factors: solar radiation and surface temperature. These elements independently play a crucial role in determining both the spatial pattern and the intensity of the UGS-BVOC emissions. Solar radiation directly influences the rate of photosynthesis and, consequently, the production of BVOCs, while temperature affects not only the physiological processes of vegetation but also the volatilization rate of these compounds (Fuhrer et al., 1997; Lombardozzi et al., 2015). The intricate interplay between these factors leads to spatial variations in the UGS-BVOC emissions, with areas receiving higher solar radiation and experiencing warmer temperatures typically exhibiting more intense BVOC emissions.

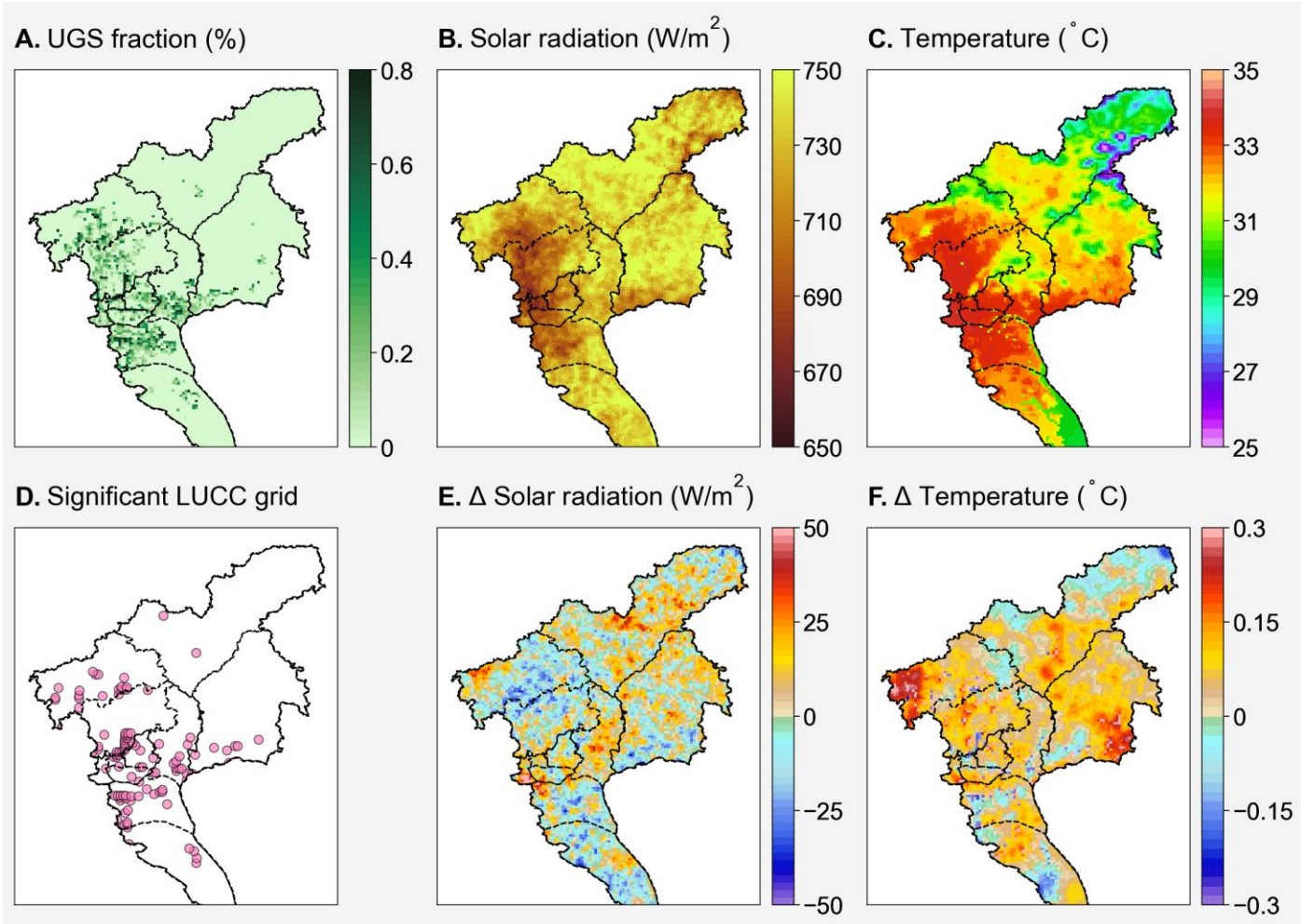

**Figure 5 The UGS map (A) and the meteorological fields during September 2017 from the Ghr_N case (B and C). (D) is the grid locations where the land use experienced significant changes, (E) and (F) are the differences in solar radiation and temperature during the analysis periods (1 September 2017 to 30 September 2017) in various land use cover data (Ghr - Gdef).**

This section has conclusively demonstrated that during the high $O_3$ season (September) in Guangzhou, the contribution of UGS-BVOC is substantial and cannot be overlooked and a notable finding is the strong spatial heterogeneity in these emissions across the city. The analysis also highlights high-resolution land use cover data increase the estimation of the UGS-BVOC emissions in the city center region.

## 3.3 Impact of UGS-LUCC and UGS-BVOC on Ozone Concentrations

The study evaluates the effects of UGS-BVOC and UGS-LUCC on MDA8 $O_3$ concentrations in Guangzhou, both individually and in combination. Figure 6 presents the absolute contributions from various cases, while the relative differences are shown in Figure S4. The analysis reveals that the UGS-BVOC emissions alone (Figure 6A) primarily affect the city center region, greatly increasing MDA8 $O_3$ concentrations by 1.0-1.4 ppb (+2.3-3.2%), which increment aligns with findings from Los Angeles, where Schlaerth et al., (2023) reported a contribution of 1.2 ppb from UGS-BVOC to urban MDA8 $O_3$ levels. N. Wang et al. (2019) reported that VOC levels can be highly sensitive in VOC-limited regions, where sufficient $NO_x$ concentrations mean that

even a small disturbance in VOCs can cause significant changes in $O_3$ concentrations. Similarly, metropolitan areas, such as Guangzhou, often experience VOC-limited conditions or $NO_x$-saturation (P. Wang et al., 2019). Consequently, the UGS-BVOC case results in an overall increase in MDA8 $O_3$. In contrast, the sole impact of the UGS-LUCC effects (Figure 6B) is more extensive, influencing both the city center and suburban regions and resulting in a general increase of approximately 1.1-2.0 ppb (+2.3-4.3%) in MDA8 $O_3$ levels, which can be attributed to the higher temperature and solar radiation (Figure 5E-F). In Guangzhou, the transformation of urban surfaces to natural vegetation due to UGS-LUCC results in lower albedo and consequently lower temperatures. However, this change also reduces the height of the urban canopy, diminishing its shading effects on solar radiation and paradoxically leading to higher temperatures in some regions. Therefore, considering the UGS-LUCC effect, the decreased urban canopy height could lead to elevated temperatures, thereby potentially increasing ozone production. However, the most significant results emerge under the combined effect of UGS-BVOC and UGS-LUCC (Figure 6C), where a substantial increase in $O_3$ concentration, ranging from 1.7-3.7 ppb (+3.8-8.5%), is observed across both the city center and suburban regions. The observed increase suggests a potentially significant influence of UGS-BVOC emissions and UGS-LUCC on ozone levels, indicating that these factors may play an important role in ozone pollution research and should be carefully considered. This finding underscores the essential role that integrated urban planning and environmental management play in controlling ozone pollution within metropolitan regions. By considering UGS-BVOC emissions in air quality models and management plans, managers can make more informed decisions to mitigate ozone levels and improve regional air quality.

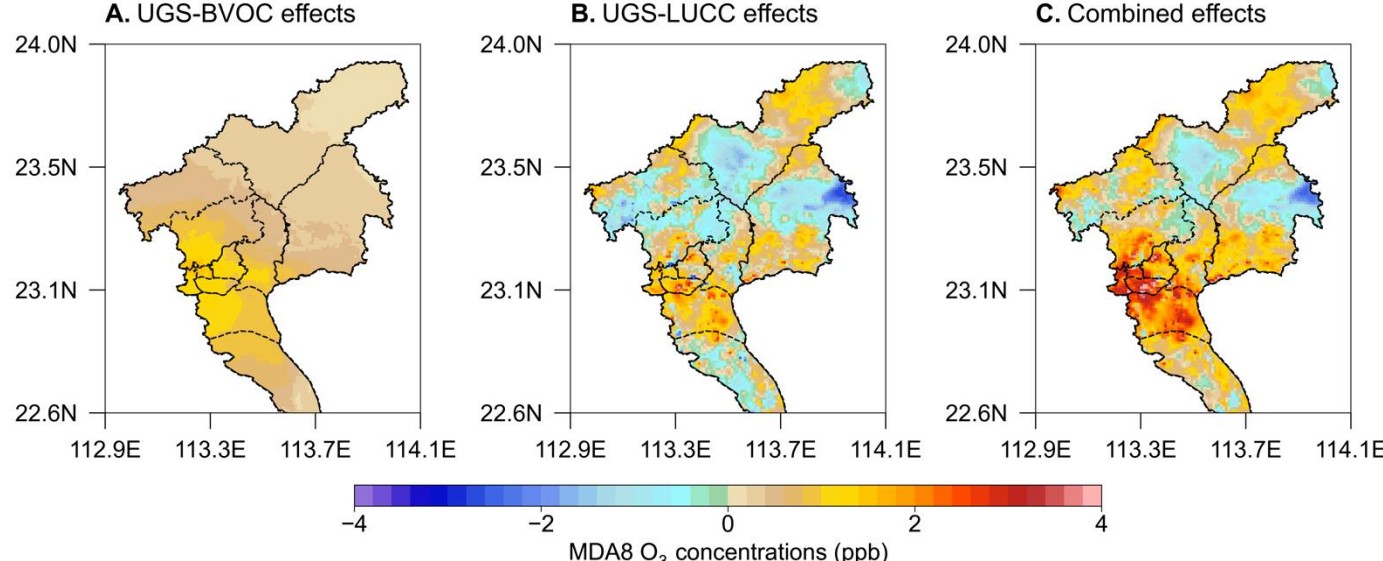

**Figure 6 The map of UGS-BVOC effects (a), LUCC effects (b), and combined effects (c) in MDA8 O3. Each map shows the difference in average MDA8 O3 concentrations for each case (Gdef_Y, Ghr_N, and Ghr_Y) relative to the Gdef_N case during September 2017.**

Previous studies have established that $O_3$ episodes are often accompanied by high temperatures and intense

solar radiation, conditions that can exacerbate the UGS-BVOC emissions, critically affecting air quality model performance (Shan et al., 2023; Soleimanian et al., 2023). In this study, an $O_3$ episode is defined as a period of two or more consecutive days with MDA8 $O_3$ concentrations exceeding 160 μg/m³ (~80 ppb) (Wu et al., 2020). Our analysis, as depicted in Figure 7A, identified two such episodes in Guangzhou City during September: the first from September 16 to 21 and the second from September 26 to 28. The Gdef_N case successfully captures these episodes but tends to underestimate both MDA8 $O_3$ during these episodes. Figure 7B-C highlight a notable reduction in wind speed during both episodes, particularly during the second episode. Despite some diffusion enhancement due to increased PBLH with rising surface temperatures (Figure S5), the surface temperature hike concurrently fosters $O_3$ production. Consequently, the episodes were dominated by a combination of temperature increases, which elevated $O_3$ concentrations, and wind speed decreases. Furthermore, Figure 7C illustrates that there was a significant spike in carbon monoxide (CO) concentrations during these episodes. CO, often used as a tracer in studies, indicates the worsening of diffusion conditions, leading to the accumulation of $NO_2$ - a primary $O_3$ precursor - thereby culminating in $O_3$ episodes in Guangzhou city.

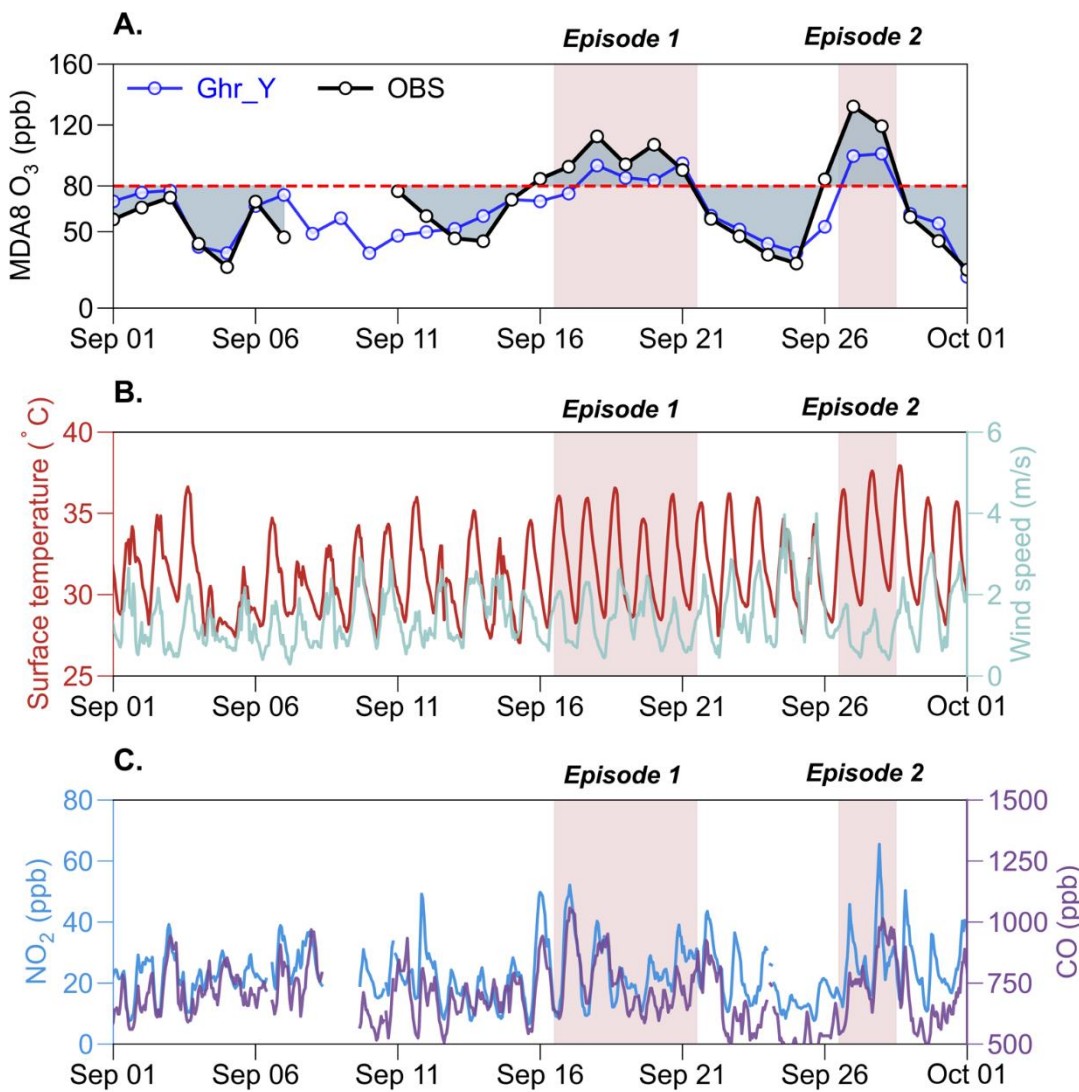

476

**Figure 7 The comparison during September 2017 between the average values from simulation results grids which have air quality stations produced by the Gdef_N case and the average observation values for MDA8 O₃ (A). (B) are the meteorological fields from the average values from the simulation result grids, which have the same locations as the air quality stations. (C) are the observed average values for NO₂ and CO concentrations from all air quality stations.**

481

Figure 8 presents that in assessing the simulation of $O_3$ episodes, the Gdef_N case, which initially underestimated $O_3$ concentrations, prompted an evaluation of improvements using different cases: UGS-LUCC (Ghr_N), UGS-BVOC emissions (Gdef_Y), and a combination of both (Ghr_Y). The analysis, focusing on the city-center, suburban, and rural stations, reveals that all cases tend to underestimate $O_3$ levels between both episodes. However, incorporating the UGS-BVOC emissions into the model results in a notable increase in mean simulated MDA8 $O_3$ concentrations, particularly in the city center and suburban regions. For the sites in the city center region, the mean simulated MDA8 $O_3$ increased by +1.6 and +2.9 ppb (+1.8% and +3.3%, respectively), while for suburban sites, the increase was +1.0 ppb (+1.1%) and +1.6 ppb (+1.9%), with rural sites experiencing a smaller increase of only +0.5 ppb (+0.5%) and +0.5 ppb (+0.7%). This trend indicates a more pronounced impact in the city center and suburban regions compared to the rural region.

Notably, the influence of the UGS-BVOC emissions on MDA8 $O_3$ in Episode 2 (+2.9 ppb) was significantly greater than in Episode 1 (+1.6 ppb), suggesting that meteorological conditions in Episode 2 were more conducive to the UGS-BVOC emissions, particularly in the city center region, which usually is VOC-limited areas.

In Episode 1, the UGS-LUCC effects on $O_3$ concentrations were comparable to that of UGS-BVOC emissions, but in Episode 2, the UGS-LUCC effects led to a near doubling of urban MDA8 $O_3$ increase by 5.9 ppb (+6.48%) compared to the UGS-BVOC emissions. This indicates that the UGS-LUCC effects play a non-negligible role in $O_3$ pollution studies, and the response to such changes under different meteorological conditions varies significantly. Furthermore, due to the limited proportion of UGS in suburban and rural areas, the increased effect of UGS on $O_3$ is less pronounced in these regions, and nearly negligible in Episode 2. While the UGS-BVOC emissions alone have a modest effect on $O_3$ concentrations, their impact can become significant when combined with the UGS-LUCC effects. For instance, the combined effects in the city center region increased by 3.2 ppb (+3.7%) and 8.9 ppb (+10.0%) during Episode 1 and Episode 2, respectively.

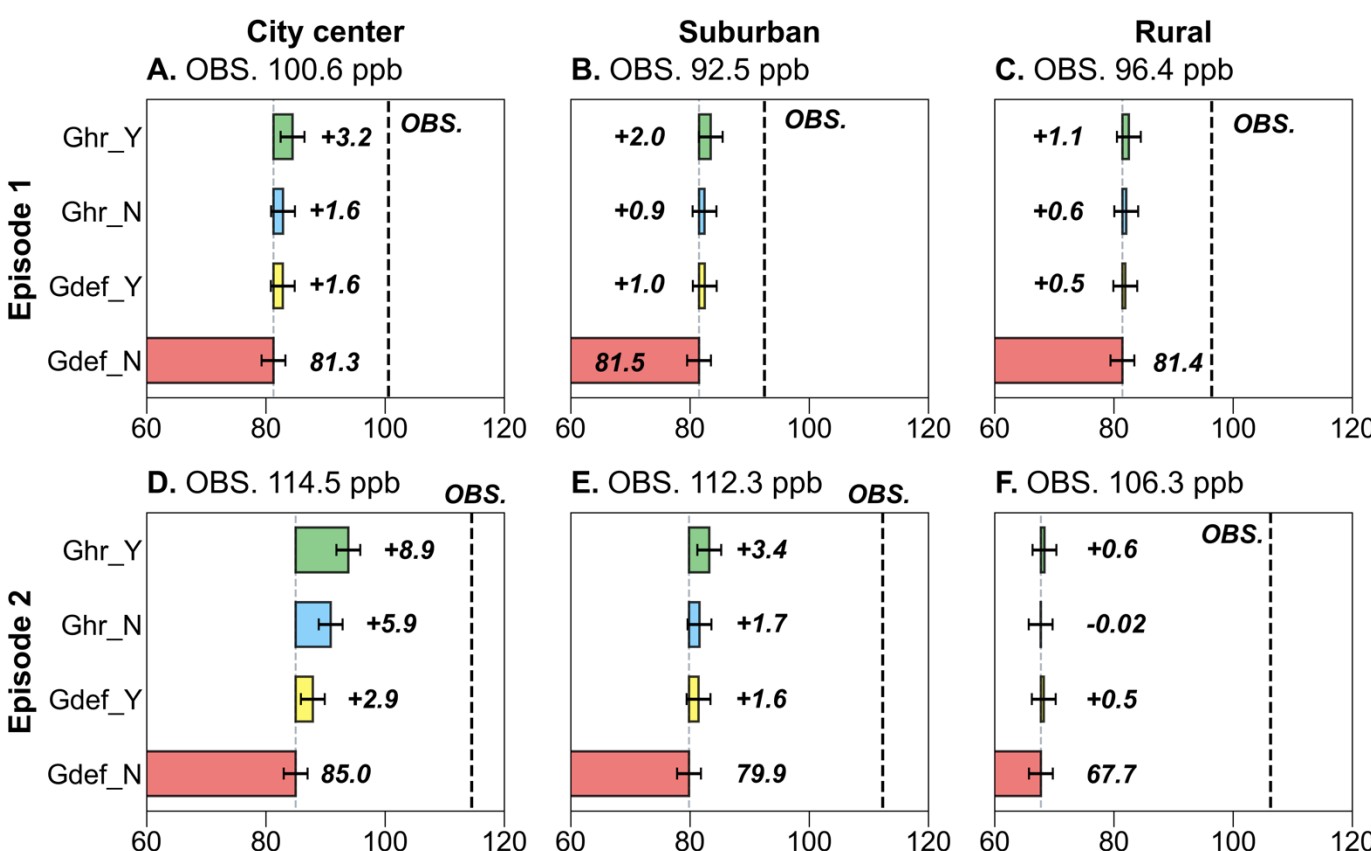

**Figure 8 Comparison of simulated versus observed mean MDA8 $O_3$ concentrations across different cases for two episodes. The figure is organized into columns representing city center (4 sites), suburban (3 sites), and rural (2 sites) settings (columns 1-3, respectively) and rows indicating comparisons for episode 1 and episode 2 (rows 1 and 2, respectively).**

Figure 9 presents the map of each effect on MDA8 $O_3$ in both episodes and the influence of UGS-BVOC

emissions (Figure 9A and Figure 9D) on the MDA8 $O_3$ concentration during Episode 1 and Episode 2 ranges

from 0 to 2.0 ppb and 0 to 3.5 ppb, respectively, with the city center region witnessing the most significant

impact. This variance can be primarily ascribed to the heightened temperatures during Episode 2 (Figure 7B),

which create conditions more conducive to ozone generation through UGS-BVOC emissions. Furthermore,

the UGS-LUCC effect's maximal contribution to the urban MDA8 $O_3$ levels could escalate to 2.2 ppb in

Episode 1 and 23.7 ppb in Episode 2 while the combined effects of UGS-LUCC and UGS-BVOC emissions

are projected to enhance MDA8 $O_3$ concentrations to 4.8 ppb and 25.2 ppb for the respective episodes. This

marked increase in the episodes' contributions can be linked to the differential responsiveness of land use data

under various meteorological conditions. Notably, while the contribution from the UGS-BVOC effect during

Episode 2 substantially exceeds that of Episode 1, the incremental impact of UGS-LUCC on combined effects

in Episode 2 is notably smaller than in Episode 1. This phenomenon indicates that the escalated UGS-BVOC

emissions in Episode 2 may start to inhibit ozone production rates incrementally.

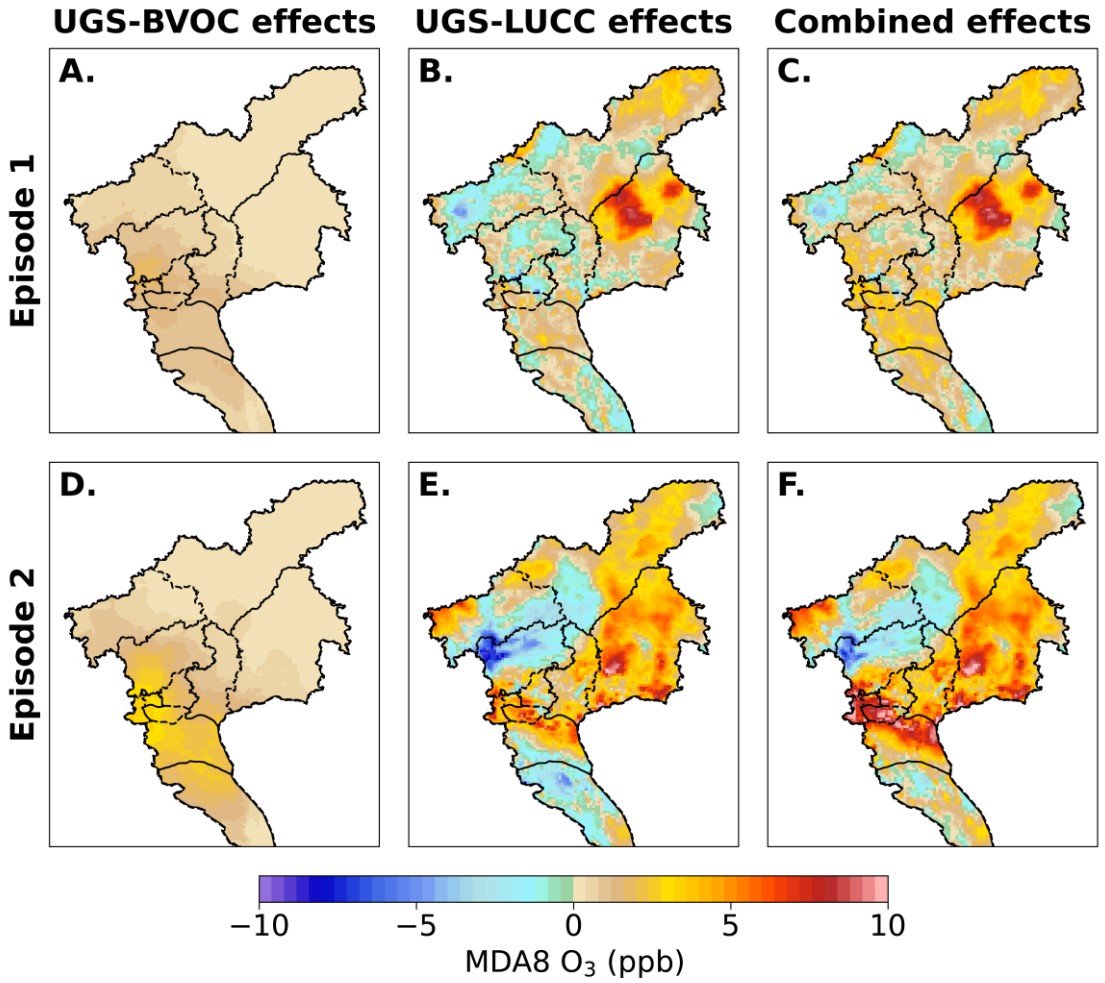

**Figure 9 The UGS-BVOC effects (A, D), the UGS-LUCC effects (B, E), and the combined effects (C, F) in Episode 1 and Episode 2, respectively.**

Figure 8 reveals that both observed $O_3$ episodes were primarily caused by reduced diffusion conditions.

Notably, the impact of the UGS-LUCC effects varied significantly between the two episodes. Analysis of meteorological variables, specifically wind speed and PBLH, which are crucial for diffusion, demonstrate distinct patterns. Figure 10 illustrates that during Episode 1, the UGS-LUCC effects led to a notable increase in the frequency of higher wind speeds (1.2–1.4 m/s) and a simultaneous decrease in the frequency of lower wind speeds (0.9–1.1 m/s). This shift in the wind speed distribution suggests an overall increase in average wind speed due to the UGS-LUCC effects during Episode 1. In contrast, Episode 2 experiences a significant decrease in wind speed frequency at 0.7-1.1 m/s, with an increase in the lower range of 0.5-0.7 m/s. This suggests that the UGS-LUCC effects further reduce the already low wind speeds in Episode 2. Concerning PBLH, the UGS-LUCC effects are observed to elevate PBLH during Episode 1, which led to a decrease in PBLH during Episode 2. Therefore, the UGS-LUCC effects are markedly more pronounced in Episode 2 than in Episode 1, contributing to a more substantial alteration of meteorological conditions affecting air dispersion and, consequently, $O_3$ formation.

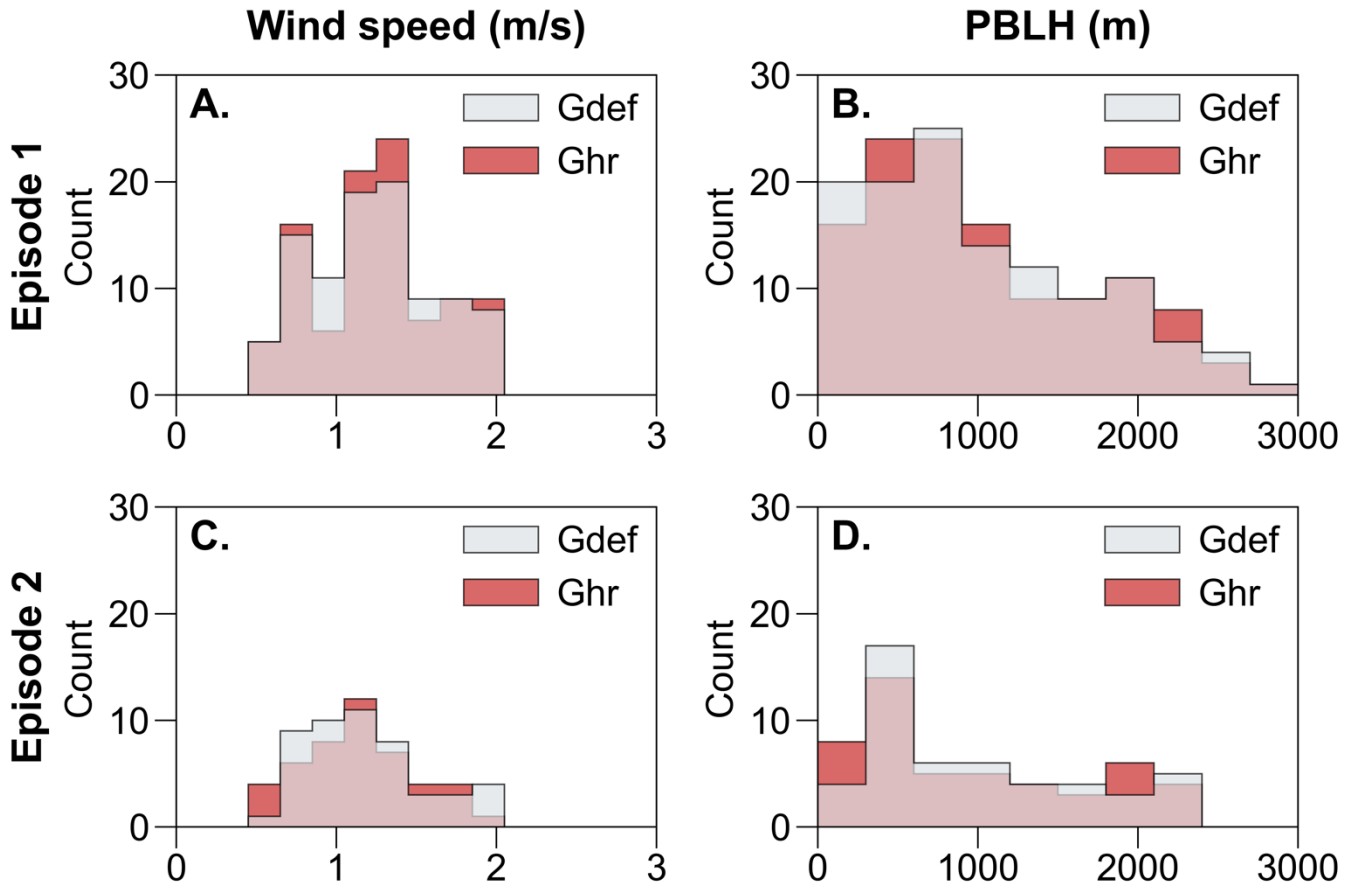

**Figure 10 The frequency of wind speed (column 1) and PBLH (column 2) in Episode 1 (row 1) and Episode 2 (row 2) driven by different land use cover datasets.**

错误!未找到引用源。 presents the overall results for the impacts of UGS-LUCC and UGS-BVOC on MDA8 $O_3$ concentrations. The effects show slight variations across different regions during September, while the effects during the two episodes exhibit more significant changes. In the city center region, which shows the

largest changes, the UGS-BVOC effect shows increases by +1.6 ppb in Episode 1 and +5.9 ppb in Episode 2, indicating that the UGS-BVOC effects influence MDA8 $O_3$ concentrations in the city center during ozone episodes, while their impact is minimal in suburban and rural regions. These results highlight the important effects of UGS-LUCC and UGS-BVOC in urban areas, especially during $O_3$ pollution periods.

**Table 6 Summary of Average MDA8 $O_3$ Concentrations (ppb) for Various Effects during September 2017.**

| Regions | Periods | UGS-BVOC effect | UGS-LUCC effect | Combined effect |
|---|---|---|---|---|
| City center | Monthly | +0.4 | 0.0 | 0.4 |
| | Episode 1 | +1.6 | +1.6 | +3.2 |
| | Episode 2 | +2.9 | +5.9 | +8.9 |
| Suburban | Monthly | +0.4 | 0.0 | 0.4 |
| | Episode 1 | +1.5 | +0.9 | +2.0 |
| | Episode 2 | +1.6 | +1.7 | +3.4 |
| Rural | Monthly | +0.4 | 0.0 | 0.4 |
| | Episode 1 | +0.5 | +0.6 | +1.1 |
| | Episode 2 | +0.5 | 0.0 | +0.5 |

# 4. Uncertainties and Limitations

In this study, we used land use and land cover data integrated at 1-km and 10-m resolutions to define the urban boundary and characterize the spatial distribution of UGS in Guangzhou. Additionally, we incorporate high-resolution LAI data, obtained through machine learning, as input for the MEGAN model. Using the WRF-CMAQ model, we quantify the effects of UGS-BVOC, UGS-LUCC, and their combined impacts on ozone concentrations in Guangzhou. However, some uncertainties and limitations remain.

First, the 10-m resolution land use and land cover data still cannot fully capture the spatial pattern of UGS in Guangzhou. As shown in Figure S2, although UGS in Guangzhou is primarily composed of EBTs, most of these EBTs are distributed along urban edges. This may result from distortions in the definition of urban extent, such as misclassifying mixed urban-vegetation grids as urban grids, caused by the coarse resolution of the 1-km land use and land cover data. The fuzzy definition of urban boundaries could lead to non-UGS areas being misclassified as UGS, potentially resulting in an overestimation of UGS-BVOC emissions.

Second, due to resolution limitations, only larger patches of grassland, cropland, and woodland are recognized as UGS, while smaller UGS vegetation, such as street trees, often goes undetected at a 10-m resolution. This omission can lead to an underestimation of the UGS-BVOC emissions.

Third, the 10-m and 1-km resolution land use and land cover data, along with the growth forms and ecotype data, use simplified categorizations for grids, which cannot fully capture the diversity of vegetation species within UGS. Since different vegetation species have varying emission factors, this simplification introduces some errors. Similarly, the oversimplified classification of land grids limits this study's ability to provide specific planning strategies for UGS at the species level. Nevertheless, it can highlight the importance of considering UGS-BVOC and UGS-LUCC in air pollution prevention and control policies.

Finally, Guangzhou, the study area, is a highly urbanized Chinese metropolis with a VOC-limited region (Gong et al., 2018; Kai et al., 2011; Liu et al., 2021). As a result, even a relatively small amount of VOC emissions, such as those from UGS-BVOC, can significantly impact ozone concentrations. Therefore, policymakers in Guangzhou should prioritize addressing the role of UGS-BVOC emissions in air pollution prevention and control. In other cities, particularly those with advanced urban development, high $NO_x$ emissions—often resulting from factors like high motor vehicle ownership—can lead to VOC-limited conditions. In such areas, it is equally important to emphasize the role of UGS-BVOC emissions in ozone pollution. In contrast, cities with lower $NO_x$ emissions identified as $NO_x$-limited regions may experience minimal impact from UGS-BVOC emissions on ozone concentrations.

## 5. Conclusion

The rapid urbanization process is accompanied by a higher frequency of ozone episodes. It has been increasingly recognized that UGS can potentially exacerbate ozone pollution under specific conditions due to the UGS-LUCC and UGS-BVOC emissions. Guangzhou, located in southern China and known as a pioneer city in reform and opening-up policies, has experienced rapid urbanization over the past thirty years, leading to increased challenges with ozone pollution. Despite efforts to reduce anthropogenic emissions, ozone episodes occur with relatively high frequency in Guangzhou. This study selected September 2017, a month with a high incidence of ozone episodes in Guangzhou, to estimate the UGS-BVOC emissions using the WRF-MEGAN model and quantitatively assess the impact of UGS-LUCC, UGS-BVOC, and their combined effects on two ozone episodes in September 2017 using the CMAQ model. The major findings are shown as follows.

1. In September 2017, the UGS-BVOC emissions in Guangzhou totaled 666 Gg, with ISOP and TERP

as the major species, emitting 213 and 136 Gg, respectively. Spatially, UGS-BVOC emissions were predominantly located in the city center region, attributed to the more extensive distribution of UGS there. The study also indicates that meteorological changes caused by UGS-LUCC do not significantly affect UGS-BVOC emissions. Instead, the formation of emission spatial distribution and intensity is closely related to local surface temperature and solar radiation. This understanding underscores the importance of considering local solar radiation and temperature conditions when assessing and modeling the distribution of the UGS-BVOC emissions, as they are pivotal in driving the spatial characteristics of these emissions.

2. Considering the UGS-BVOC and UGS-LUCC effects can effectively mitigate the underestimation of surface ozone concentrations by regional air quality models, though other factors such as inaccuracies in emissions inventories, chemical mechanisms, and meteorological inputs may also contribute to these underestimations. For instance, incorporating UGS-BVOC emissions results in an increase in ISOP concentration from 0.29 ppb to 0.35 ppb and from 0.23 ppb to 0.29 ppb under different land use cases (Gdef and Ghr), compared to a baseline concentration of 0.35 ppb. This significant enhancement in ISOP concentrations—the predominant component in BVOCs and the most crucial VOC for $O_3$ formation in the PRD—highlights two key points. Firstly, it indicates an improvement in the accuracy of BVOC concentration simulations. Secondly, this precise estimation of BVOCs has notably shifted the MB of $O_3$ simulations from 3.62 ppb to -0.75 ppb in the city center reigon. Additionally, the simulation of $NO_2$ concentrations also shows slight improvements, with the MB decreasing from 3.27 ppb to 2.81 ppb upon accounting for UGS-BVOCs and UGS-LUCC. Given that the UGS are often located in densely populated urban regions, their inclusion in air quality simulations is crucial for accurately modeling urban air quality.

3. The UGS-BVOC emissions have a remarkable impact on ozone concentrations, with increases ranging from 1.0-1.4 ppb (+2.3-3.2%) in the city center regions. However, when considering the combined UGS-LUCC and UGS-BVOC effects, the impact on MDA8 $O_3$ concentrations becomes remarkable, with values ranging from 1.7-3.7 ppb (+3.8-8.5%) in the city center region. This indicates the importance of considering both UGS-LUCC and UGS-BVOC impacts when discussing the influence of UGS on air quality. Since UGS exhibits different effects in various ozone episodes, it is found that the impact of UGS on ozone levels is related to specific meteorological conditions. In the episodes of this study, the combined effects on MDA8 $O_3$ can reach up to 8.9 ppb in the city center region.

This study on ozone pollution in Guangzhou provides key insights for other cities on integrating UGS with

air quality management. By including UGS-BVOC emissions and UGS-LUCC in the air quality model, the study demonstrates improved accuracy in predicting surface ozone concentrations, which can aid urban planners and environmental policymakers in refining their strategies to better address urban air pollution. Moreover, these findings encourage cities to integrate urban forestry into their land use planning and air quality frameworks, promoting environmental sustainability amid rapid urbanization.

## Data availability

The WRF (Weather Research and Forecasting Model) code can be obtained from the official repository at https://github.com/wrf-model/WRF. The CMAQ (Community Multiscale Air Quality Model) code is accessible at https://github.com/USEPA/CMAQ. Model output data used for analysis and plotting, and the code used for simulations can be made available upon request (Haofan Wang, wanghf58@mail2.sysu.edu.cn).

## Author contributions

HFW conceived the study, carried out the model simulations, and drafted the manuscript. YJL completed the data visualization. YML conceived and supervised this study, and reviewed and edited the paper. XL and YZ provided useful comments on the paper. QF supervised and funded the study. CS provided the meteorological data for model evaluation. SCL, YZ, TZ, and DLY provided the observation data for the evaluation of isoprene simulation.

## Competing interests

The contact author has declared that neither they nor their co-authors have any competing interests.

## Acknowledgment

The authors would like to thank the MEIC team from Tsinghua University for providing the anthropogenic emissions for this study. Meanwhile, we also thank Tianhe-2 for providing the computer resources.

## Financial support

This work is jointly funded by the innovation Group Project of Southern Marine Science and Engineering Guangdong Laboratory (Zhuhai) (316323005), Guangdong Basic and Applied Basic Research Foundation (2020B0301030004, 2023A1515110103, 2023A1515010162), National Natural Science Foundation of China (42105097, 42075181, 42375182), Guangdong Science and Technology Planning Project (2019B121201002), and Science and Technology Planning Project of Guangzhou (2023A04J1544).

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
