# Peer review of "Underappreciated contributions of biogenic volatile organic compounds from urban"

_EGUsphere, 2024_

## Author Comment (AC1)

**Response to Reviewer #3:**

We gratefully thank the editor and all reviewers for their time spent making their constructive remarks and useful suggestions, which have significantly raised the quality of the manuscript and have enabled us to improve the manuscript. Each suggested revision and comment, brought forward by the reviewers was accurately incorporated and considered. Below are the comments of the reviewers and response point by point and the revisions are indicated. We use different colored fonts to distinguish between responses to reviewers and the revised sections of the manuscript.

1. Responses to reviewers are highlighted in blue.

2. *Revised sections of the manuscript are highlighted in red.*

**Comment 1**: This paper uses high-resolution WRF-CMAQ simulations to investigate the impacts of urban greening in Guangzhou, China through the use of different resolution input datasets on land cover and leaf area indices. The impact of these different model configurations is usefully explored through comparisons with observations. Representing urban green spaces in models is important for understanding ozone atmospheric chemistry in urban settings. This study is relevant to ACP. I recommend publication of this manuscript after the concerns specified below have been addressed.

**Reply:** We sincerely thank the reviewer for the thoughtful and encouraging feedback on our study. We are pleased that you recognize the relevance of our work in representing urban green spaces and investigating ozone atmospheric chemistry in urban settings. Your acknowledgment of the importance of high-resolution simulations and comparisons with observations is greatly appreciated. We will carefully address the concerns raised to improve the manuscript and ensure it meets the standards of ACP. Thank you for your constructive feedback and support.

**Comment 2**: The manuscript discusses the use of different resolution input datasets and uses a high resolution, nested model. It would be useful if the authors could include a section on the benefits and limitations of using high resolution. For example, you have 1 km versus 10 m land cover data, but when running the model, you have all of this information in a 1 km grid box. How do these high-resolution datasets impact the calculations of BVOCs in the model? What are the uncertainties here?

**Reply:** Thanks for the valuable suggestion. We have added a "Uncertainties and Limitations" section to explain it.

*First, the 10-m resolution land use and land cover data still cannot fully capture the spatial pattern of UGS in Guangzhou. As shown in Figure S2, although UGS in Guangzhou is primarily composed of EBTs, most of these EBTs are distributed along urban edges. This may result from distortions in the definition of urban extent,*

*such as misclassifying mixed urban-vegetation grids as urban grids, caused by the coarse resolution of the 1-km land use and land cover data. The fuzzy definition of urban boundaries could lead to non-UGS areas being misclassified as UGS, potentially resulting in an overestimation of UGS-BVOC emissions.*

**Comment 3**: Although the authors compare the simulations to observations of meteorological parameters, $O_3$, $NO_2$, and isoprene, there is very little discussion on how these observations were measured and the time resolution. It would be helpful to readers to elaborate more on this.

**Reply:** Thanks for the valuable suggestion. We have added some information about the measurement in Section 2.4.

*The real-time hourly concentration of $O_3$ was measured by the ultraviolet absorption spectrometry method and differential optical absorption spectroscopy at each monitoring site. $NO_2$ concentrations are measured by the molybdenum converter method known to have positive interferences from $NO_2$ oxidation products (Dunlea et al., 2007).The instrumental operation, maintenance, data assurance, and quality control were properly conducted based on the most recent revisions of China Environmental Protection Standards (Zhang and Cao, 2015), and the locations of these air quality stations are depicted in Figure 1. Additionally, meteorological data also undergo thorough quality control. Subsequently, they are utilized to assess the model performance of WRF-CMAQ.*

*For the isoprene (ISOP) evaluation, we use observation data from the Modiesha (23.11°N, 113.33°E) and Wanqingsha (22.71°N, 113.55°E) sites (Figure 1), where an online gas chromatography-mass spectrometry/flame ionization detector system (GC-FID/MSD, TH 300B, Wuhan) is used to measure VOCs in the ambient atmosphere. The system has a sampling rate of 60 mL/min for 5 minutes per sample, with a sampling frequency of once per hour (Meng et al., 2022). The ISOP observation data undergo rigorous quality control, which can be used for evaluating simulated ISOP concentrations. It is worth noting that the ISOP observational data for the Modiesha site covers September 2017, while the Wanqingsha site has data coverage from September 7 to September 30, 2017.*

**Comment 4**: In this manuscript, the authors are addressing the sensitivity of $O_3$ to changes in BVOCs, but there is little discussion on whether $O_3$ is NOx-limited or VOC-limited in Guangzhou. There is also no discussion on other potential sources affecting the area. The authors mention that there is rapid urbanization happening in Guangzhou. Does this mean that there is more industry, more vehicles, etc. that could be impacting $O_3$ production and loss processes? $O_3$ is nonlinear, and depending on the regime, and other factors like emissions, the composition and ratios of VOCs and $NO_x$ in the area will affect $O_3$ differently.

**Reply:** Thanks for the very valuable suggestions. We have discussed the VOC-limited or NOx-limited conditions of Guangzhou in Section 3.3 and 4.

*Section 3.3:*

*N. Wang et al. (2019) reported that VOC levels can be highly sensitive in VOC-limited regions, where sufficient $NO_x$ concentrations mean that even a small disturbance in VOCs can cause significant changes in $O_3$ concentrations. Similarly, metropolitan areas, such as Guangzhou, often experience VOC-limited conditions or $NO_x$-saturation (P. Wang et al., 2019). Consequently, the UGS-BVOC case results in an overall increase in MDA8 $O_3$.*

*Section 4:*

*Finally, Guangzhou, the study area, is a highly urbanized Chinese metropolis with a VOC-limited region (Gong et al., 2018; Kai et al., 2011; Liu et al., 2021). As a result, even a relatively small amount of VOC emissions, such as those from UGS-BVOC, can significantly impact ozone concentrations. Therefore, policymakers in Guangzhou should prioritize addressing the role of UGS-BVOC emissions in air pollution prevention and control. In other cities, particularly those with advanced urban development, high $NO_x$ emissions—often resulting from factors like high motor vehicle ownership—can lead to VOC-limited conditions. In such areas, it is equally important to emphasize the role of UGS-BVOC emissions in ozone pollution. In contrast, cities with lower $NO_x$ emissions identified as $NO_x$-limited regions may experience minimal impact from UGS-BVOC emissions on ozone concentrations.*

**Comment 5**: There is an inconsistency with the units, which can be confusing to readers. At times, the authors use "ppb" and at other times they use "$\mu g/m^3$" for gases. Please use a consistent unit throughout the paper.

**Reply:** Thanks for the valuable suggestion. We have changed all the unit "$\mu g/m^3$" to "ppb" for gases.

**Comment 6**: For the tables and figures in the paper, please specify the timeframes that the concentrations are averaged over.

**Reply:** Thanks for the nice suggestions. We have added the timeframes in the titles of tables and figures.

**Comment 7**: Figure 1: This is a paper based on Guangzhou. For readers who might not be familiar with the area, could the authors provide more description on the urban, suburban, and rural regions? Is there much vegetation in the urban centers?

**Reply:** Thanks for the valuable suggestion. We have added some information about the city center, suburban, and rural regions.

*The city center region has more UGS areas due to the higher urban land use and land cover fraction (Figure S1) compared to the suburban and rural regions.*

**Comment 8**: Section 2.2: Did the authors use online or offline MEGAN? It seems to me that you are referring to MEGANv2.1, not v3.1. Please clarify, as the resolution of the model can have impacts on online emission calculations.

**Reply:** We used the online MEGAN in the CMAQ simulation. We have updated the Section 2.2 and the updated version is for the MEGANv3.1.

*In this equation, E is the net emission flux ($\mu g \ m^{-2} \ h^{-1}$), and EF is the weighted average of the emission factor ($\mu g \ m^{-2} \ h^{-1}$) for each vegetation type calculated by Emission Factor Processor (EFP). The emission activity factor ($\gamma$) considers emission responses to changes in environmental and phenological conditions. Compare with earlier versions, $\gamma$ in MEGANv3.1 adds quantifications for responses to high and low temperature, high wind speed, and air pollution ($O_3$).*

$$\gamma = LAI \times \gamma_{TP} \times \gamma_{LA} \times \gamma_{SM} \times \gamma_{HT} \times \gamma_{LT} \times \gamma_{HW} \times \gamma_{CO_2} \times \gamma_{BD} \times \gamma_{O_3} \qquad (Eq.\ 2)$$

*In this equation, the activity factor denotes the emission response to canopy temperature/light($\gamma_{TP}$), leaf age ($\gamma_{LA}$), soil moisture ($\gamma_{SM}$), high temperature ($\gamma_{HT}$), low temperature ($\gamma_{LT}$), high wind speed ($\gamma_{HW}$), ambient $CO_2$ concentration ($\gamma_{CO_2}$), bidirectional exchange ($\gamma_{BD}$), $O_3$ exposure ($\gamma_{O_3}$), and Leaf Area Index (LAI). In this study, $\gamma_{CO_2}$ was not considered in the BVOC emission estimation. The MEGANv3.1 approach can calculate the emissions at each canopy level as the product of the emission factor and emission activity at each level.*

**Comment 9**: Section 2.3 and onwards: I would recommend that the authors refrain from using the term "scenario" to describe the different simulations. These are model configurations, whereas the term "scenario" tends to highlight a potential future or possible outcomes.

**Reply:** Thanks for the very nice suggestions. We have replaced the "scenario" to "case" in the whole manuscript.

**Comment 10**: Section 2.4: What kind of instrumentation was used to measure $O_3$?

**Reply:** Thanks for the very carefully checking. We have added the description of the instrumentation for measuring $O_3$ and $NO_2$.

*The real-time hourly concentration of $O_3$ was measured by the ultraviolet absorption spectrometry method and differential optical absorption spectroscopy at each monitoring site. $NO_2$ concentrations are measured by*

*the molybdenum converter method known to have positive interferences from NO₂ oxidation products (Dunlea et al., 2007).The instrumental operation, maintenance, data assurance, and quality control were properly conducted based on the most recent revisions of China Environmental Protection Standards (Zhang and Cao, 2015), and the locations of these air quality stations are depicted in Figure 1. Additionally, meteorological data also undergo thorough quality control. Subsequently, they are utilized to assess the model performance of WRF-CMAQ.*

**Comment 11**: Table 2: Are these monthly concentrations? Please clarify in the text or in the caption.

**Reply:** Thanks for this suggestion. We have reorganized this table.

*Table 1 The evaluation results for the monthly mean ISOP concentrations. The "Gdef_N", "Gdef_Y", "Ghr_N", and "Ghr_Y" columns show the various metrics from comparing the hourly observation and simulation values during September 2017 for the Modiesha site and 7 September 2017 to 30 September 2017 for the Wanqingsha site.*

| Site name | Metrics | Gdef_N (ppb) | Gdef_Y (ppb) | Ghr_N (ppb) | Ghr_Y (ppb) |
|-----------|---------|--------------|--------------|-------------|-------------|
| Modiesha | Sim. | 0.29 | 0.35 | 0.23 | 0.29 |
| | Obs. | 0.34 | 0.34 | 0.34 | 0.34 |
| | MB | -0.06 | 0.01 | -0.11 | -0.05 |
| | NME | 76.0% | 68.7% | 73.6% | 66.2% |
| | NMB | -16.4% | 3.5% | -31.3% | -13.1% |
| | R | 0.44 | 0.46 | 0.37 | 0.39 |
| Wanqingsha | Sim. | 0.29 | 0.31 | 0.27 | 0.29 |
| | Obs. | 0.45 | 0.45 | 0.45 | 0.45 |
| | MB | -0.15 | -0.14 | -0.17 | -0.15 |
| | NME | 58.9% | 56.8% | 60.4% | 58.1% |
| | NMB | -34.7% | -30.6% | -38.7% | -34.8% |
| | R | 0.35 | 0.39 | 0.34 | 0.4 |

**Comment 12**: Line 260: By "specified requirement", do the authors mean the observed values? If so, I would refer to them as that because the authors are not referring to a standard of sorts.

**Reply:** Thanks for the good suggestion. We have rewritten this sentence.

*Additionally, various statistical metrics were used to assess the performance of hourly O₃, MDA8 O₃, and NO₂ concentrations from the CMAQ simulation (Emery et al. 2017). These metrics comprise the correlation coefficient (R), normalized mean bias (NMB), and normalized mean error (NME). The formulas for these metrics are listed in Table S3. As shown in Table 3, the modeling performance for all cases are reasonably, albeit with some degree of underestimation.*

**Comment 13**: Line 265-267: The authors mention that integrating UGS-BVOC can improve the accuracy of NO₂ predictions. Can the authors elaborate on why this improvement in simulated NO₂ happens?

**Reply:** Thanks for the valuable suggestion. We have added the elaborate on why this improvement in simulated NO₂ happens.

*The improvement in NO₂ predictions is attributed to the increased involvement of NO₂ in O₃ formation caused by the UGS-BVOC emissions, which reduces simulated NO₂ concentrations and narrows its bias against the observation.*

**Comment 14**: Table 3: Again, what time frame are these concentrations averaged over? Monthly?

**Reply:** Thanks for the valuable suggestion. We have added the timeframe to the title of the table.

*Table 2 Evaluation results of the simulated monthly mean hourly O₃, MDA8 O₃, and hourly NO₂ mixing ratios for each case during September 2017.*

| Pollutant | Case name | Sim (ppb) | Obs (ppb) | MB (ppb) | NMB | NME | R |
|---|---|---|---|---|---|---|---|
| Hourly O₃ | Gdef_N | 28.23 | 30.49 | -2.26 | -6.7% | 23.6% | 0.82 |
| | Gdef_Y | 28.67 | 30.49 | -1.82 | -5.3% | 23.6% | 0.82 |
| | Ghr_N | 28.89 | 30.49 | -1.60 | -4.8% | 22.5% | 0.83 |
| | Ghr_Y | 29.33 | 30.49 | -1.15 | -3.4% | 22.4% | 0.83 |
| MDA8 O₃ | Gdef_N | 60.11 | 62.27 | -2.16 | -3.47% | 21.71% | 0.84 |
| | Gdef_Y | 61.04 | 62.27 | -1.23 | -1.97% | 21.40% | 0.84 |
| | Ghr_N | 61.07 | 62.27 | -1.20 | -1.92% | 21.28% | 0.84 |
| | Ghr_Y | 62.00 | 62.27 | -0.26 | -0.42% | 21.23% | 0.84 |
| Hourly NO₂ | Gdef_N | 24.78 | 21.50 | 3.27 | 15.2% | 45.7% | 0.63 |
| | Gdef_Y | 24.74 | 21.50 | 3.24 | 15.0% | 45.5% | 0.63 |
| | Ghr_N | 24.35 | 21.50 | 2.84 | 13.2% | 43.8% | 0.63 |
| | Ghr_Y | 24.32 | 21.50 | 2.81 | 13.0% | 43.6% | 0.63 |

**Comment 15**: Table 4: Same comment as tables 2 and 3.

**Reply:** Thanks for the valuable suggestion. We have added the timeframe to the title of the table.

*Table 3 Evaluation results of simulated monthly mean hourly O3 and MDA8 O3 mixing ratios in city center, suburban, and rural areas for each case during September 2017.*

| Variable | Regions | MB (ppb) | | | | R | | | |
|---|---|---|---|---|---|---|---|---|---|
| | | Gdef_N | Gdef_Y | Ghr_N | Ghr_Y | Gdef_N | Gdef_Y | Ghr_N | Ghr_Y |
| MDA8 O₃ | City center | −3.627 | −2.241 | −2.110 | −0.747 | 0.805 | 0.810 | 0.810 | 0.813 |
| | Suburban | −4.076 | −3.251 | −3.210 | −2.376 | 0.737 | 0.743 | 0.717 | 0.727 |
| | Rural | −5.109 | −4.757 | −4.866 | −4.528 | 0.665 | 0.655 | 0.695 | 0.690 |
| Hourly O₃ | City center | -2.862 | -2.292 | -2.086 | -1.520 | 0.800 | 0.802 | 0.811 | 0.812 |
| | Suburban | -3.148 | -2.803 | -2.647 | -2.295 | 0.824 | 0.825 | 0.824 | 0.826 |
| | Rural | -1.184 | -1.630 | -1.375 | -1.164 | 0.742 | 0.741 | 0.751 | 0.750 |

**Comment 16**: Line 288: Do you mean "monoterpene" instead of "monoethylene"?

**Reply:** Thanks for the carefully checking. We have changed the "monoethylene" to "monoterpene".

**Comment 17**: Line 473-475: This sentence is rather confusing. Can you clarify how an increase and simultaneous decrease leads to an overall increase in wind speed?

**Reply:** Thanks for this suggestion. We have rewritten this sentence.

*Figure 10 illustrates that during Episode 1, the UGS-LUCC effects led to a notable increase in the frequency of higher wind speeds (1.2–1.4 m/s) and a simultaneous decrease in the frequency of lower wind speeds (0.9–1.1 m/s). This shift in the wind speed distribution suggests an overall increase in average wind speed due to the UGS-LUCC effects during Episode 1.*

**Comment 18**: Technical Correction 1: At the beginning of the paper, the authors establish an abbreviation convention, such as TERP for monoterpenes, ISOP for isoprene, LUCC for land use cover change, etc. Throughout the paper, I found that the abbreviations were being described again on several occasions (i.e., line 288, line 277-278). Please make this consistent throughout the paper.

**Reply:** Thanks for this carefully checking. We have corrected these errors.

**Comment 19**: Technical Correction 2: When naming figures throughout the paper, the authors should use the naming convention of "Figure 1A and 1B" rather than "Figure 1 (A) and (B)".

**Reply:** Thanks for this nice suggestion. We have changed the naming convention of the figures in the manuscript.

**Comment 20**: Line 74: "the complexity of these interactions, and they demonstrated that vegetation could exert nonlinear effects on" – Remove "they" from sentence.

**Reply:** Thanks for this suggestion. We have removed "they" here.

*Furthermore, Seinfeld et al., (1998) underscores the complexity of these interactions, and demonstrated that vegetation could exert nonlinear effects on meteorological processes.*

**Comment 21**: Line 97: Should be "investigated" instead of "investigating".

**Reply:** Thanks for this suggestion. We have changed the "investigating" to "investigated".

**Comment 22**: Line 210: Space between "Table 1" and "were".

**Reply:** Thanks for this suggestion. We have added a space between "Table 1" and "were".

---

## Author Comment (AC2)

**Response to Reviewer #1:**

We gratefully thank the editor and all reviewers for their time spent making their constructive remarks and useful suggestions, which have significantly raised the quality of the manuscript and have enabled us to improve the manuscript. Each suggested revision and comment, brought forward by the reviewers was accurately incorporated and considered. Below are the comments of the reviewers and response point by point and the revisions are indicated. We use different colored fonts to distinguish between responses to reviewers and the revised sections of the manuscript.

1. Responses to reviewers are highlighted in blue.

2. *Revised sections of the manuscript are highlighted in red.*

**Comment 1**: This paper usefully demonstrates the impacts of updating biogenic VOCs in urban environments (called here UGS-BVOC) as well as updating land use cover data to high resolution maps (called here UGS-LUCC). The impact of each of these changes separately and together on ozone in Guangzhou, China is thoroughly explored. These emissions and model improvements are clearly important for accurately simulating ozone in this region. I recommend publication after major revisions specified below.

**Reply:** We sincerely thank the reviewer for their positive assessment of our study and for recognizing the importance of updating biogenic VOC emissions (UGS-BVOC) and high-resolution land use cover data (UGS-LUCC) in accurately simulating ozone in urban environments. We are pleased that you found our exploration of their separate and combined impacts on ozone in Guangzhou to be thorough and valuable. We will carefully address the specific concerns raised to further improve the manuscript and meet the standards required for publication. Thank you again for your constructive feedback and support.

**Comment 2**: While VOCs are important for ozone production, so are nitrogen oxides. This paper could be mis-leading to policy makers if $NO_x$ is not also mentioned as an ozone precursor more clearly in the text.

**Reply:** Thanks for this valuable suggestion. We have demonstrated the $O_3$ formation from $NO_x$ and VOCs in the "Introduction" Section and added some discussion about the impact of $NO_x$ levels to the "Uncertainties and Limitations" (Section 4).

*Introduction:*

*Surface $O_3$ is generally formed through chemical reactions of VOCs and $NO_x$ in the presence of sunlight.*

*Section 4:*

*Finally, Guangzhou, the study area, is a highly urbanized Chinese metropolis with a VOC-limited region (Gong et al., 2018; Kai et al., 2011; Liu et al., 2021). As a result, even a relatively small amount of VOC*

*emissions, such as those from UGS-BVOC, can significantly impact ozone concentrations. Therefore, policymakers in Guangzhou should prioritize addressing the role of UGS-BVOC emissions in air pollution prevention and control. In other cities, particularly those with advanced urban development, high $NO_x$ emissions—often resulting from factors like high motor vehicle ownership—can lead to VOC-limited conditions. In such areas, it is equally important to emphasize the role of UGS-BVOC emissions in ozone pollution. In contrast, cities with lower $NO_x$ emissions identified as $NO_x$-limited regions may experience minimal impact from UGS-BVOC emissions on ozone concentrations.*

**Comment 3**: Why does the abstract and throughout the text use ug/m³ as the units for gases in the atmosphere. The convention is typically ppb in atmospheric chemistry. Please provide a valid reason or use the conventional unit. Additionally, throughout the paper the authors switch from ug/m³ and ppb throughout the text and are not consistent with one unit. This is confusing for the reader. I would highly recommend switching all units to ppb for all gases.

**Reply:** Thank you for the nice suggestion, and we have converted all units to ppb for all gases.

**Comment 4**: The use of urban as both an "urban region" in Figure 1 and "urban landcover type" to mean two different things makes the paper hard to understand at first. This paper selects locations classified with the urban landcover type as urban and this is called UGS (urban green space), but then also classifies by region into 3 types called urban, suburban, and rural. Providing more detail on how the regions are determined in Figure 1 and lines 202 – 206 would be useful. And please consider calling this something other than "urban", so that readers can more easily differentiate when the authors are referring to urban as a region and or as a landcover type. Perhaps, terminology like "city center", "suburban", and "rural" could work.

**Reply:** Thank you for your helpful suggestion. We have revised the region definition by replacing "urban" with "city center" as recommended.

**Comment 5**: There is very little comparison to isoprene observations even though this is the major development in this paper (Table 2 and line 235). It is an excellent opportunity that the model can be compared against these two monitoring sites for isoprene. For reader, clarity, can you please add these two monitoring sites to Figure 1? Looking at the average isoprene concentrations over the entire campaign (Sept – Nov) is not sufficient proof that the isoprene emissions have improved. Also I am confused on the time periods. The observations appear to be for Sept 20 – Nov 20, but in the methods section you state the model is only run for September 1 – Sept 30 (ignoring spin-up). Are you comparing the observations and model during the same time period? It is very important to compare the model and observations over the exact same time period since isoprene emissions are very seasonally dependent. Can you provide additional statistics like you do for the

meteorological variables in Table S2 and ozone and NO₂ in Table 3? Further evaluation of the diurnal cycle in the model compared to the observations and statistics for the different months would also be very useful in understanding whether the isoprene emissions have improved in the model. Since your main updates are isoprene emissions it is very important that this evaluation is clear on what has been done and that this is a good evaluation to ensure that the updates the author has made are robust.

Can you also provide more information on how the observations were collected at these monitoring sites? What instrument technique? Were any interferences considered if the instrument technique was a PTR (Coggon, et al., 2024 - https://doi.org/10.5194/amt-17-801-2024)? Was the diurnal cycle of the observations consistent with known chemistry of isoprene where isoprene concentrations rise during the day and fall rapidly at night in high NOₓ urban locations with significant amounts of NO₃ radical? This can help you verify that any interferences for isoprene in your instrument technique are appropriately accounted for.

**Reply:** Thanks for this comment and we have added the ISOP monitoring sites to Figure 1.

[Figure]

*Figure 1 The innermost domain of WRF-CMAQ with various areas and the air quality station locations map. Modiesha and Wanqingsha are the observation sites for isoprene.*

In this revised manuscript, thanks to the contribution of Prof. Senchao Lai's groups, we have got the hourly isoprene observations at Modiesha and Wanqingsha sites for a more detailed validation. We have listed his group members as co-authors of this revised study. The observed isoprene concentration shows a reasonably feature that is high during the day and low at night. We have compared the observation and model simulation

during the whole September, which have the same time period as the model simulation. Below please find the revised validation results for isoprene.

*BVOCs are the major sources of ISOP and monoterpene (TERP), rendering the assessment of their concentrations a pivotal method for indirectly verifying the accuracy of BVOC emission estimates. Table 2 delineated within this study presents the mean concentrations of ISOP derived from various cases juxtaposed with the observed average concentrations. This comparative analysis in the Modiesha site reveals that after the incorporation of the UGS-BVOC emissions, there is an augmentation in the ISOP concentration from 0.29 to 0.35 ppb and from 0.23 to 0.29 ppb under distinct land use cover cases (Gdef and Ghr), relative to an observed concentration of 0.34 ppb. Meanwhile, the evaluation at the Wanqingsha site, where the observed mean ISOP concentration was 0.45 ppb from September 7 to September 30, 2017, shows that the modeled ISOP concentrations increased from 0.29 to 0.31 ppb and from 0.27 to 0.29 ppb under distinct land use cover cases (Gdef and Ghr) when UGS-BVOC emissions were included. This increment signifies a substantial diminution in the discrepancy between the modeled and observed concentrations attributable to the UGS-BVOC emissions. Analogously, the integration of the UGS-BVOC emissions yields a refinement in the estimation accuracy of ISOP concentrations at the Modiesha site, as evidenced by a reduced bias.*

*These findings reveal that ISOP concentrations are underestimated by 16.4% and 34.7% in the Modiesha and Wanqingsha sites when UGS-BVOCs are excluded, respectively, suggesting the important role of UGS-BVOCs emissions in modeling. Moreover, numerous studies highlight the significant role of ISOP in $O_3$ formation within the Pearl River Delta (PRD) region, including Guangzhou. For instance, Zheng et al., (2009) demonstrated that ISOP has the highest ozone formation potential among all VOCs. Therefore, incorporating UGS-BVOCs into ISOP concentration estimates is crucial for accurately modeling regional $O_3$ levels.*

*Table 1 The evaluation results for the monthly mean ISOP concentrations. The "Gdef_N", "Gdef_Y", "Ghr_N", and "Ghr_Y" columns show the various metrics from comparing the hourly observation and simulation values during September 2017 for the Modiesha site and 7 September 2017 to 30 September 2017 for the Wanqingsha site.*

| Site name | Metrics | Gdef_N (ppb) | Gdef_Y (ppb) | Ghr_N (ppb) | Ghr_Y (ppb) |
|---|---|---|---|---|---|
| Modiesha | Sim. | 0.29 | 0.35 | 0.23 | 0.29 |
| | Obs. | 0.34 | 0.34 | 0.34 | 0.34 |
| | MB | -0.06 | 0.01 | -0.11 | -0.05 |
| | NME | 76.0% | 68.7% | 73.6% | 66.2% |
| | NMB | -16.4% | 3.5% | -31.3% | -13.1% |
| | R | 0.44 | 0.46 | 0.37 | 0.39 |
| Wanqingsha | Sim. | 0.29 | 0.31 | 0.27 | 0.29 |
| | Obs. | 0.45 | 0.45 | 0.45 | 0.45 |
| | MB | -0.15 | -0.14 | -0.17 | -0.15 |
| | NME | 58.9% | 56.8% | 60.4% | 58.1% |
| | NMB | -34.7% | -30.6% | -38.7% | -34.8% |
| | R | 0.35 | 0.39 | 0.34 | 0.4 |

In the revised version, we have added the description of the measurement of ISOP.

*For the isoprene (ISOP) evaluation, we use observation data from the Modiesha (23.11°N, 113.33°E) and Wanqingsha (22.71°N, 113.55°E) sites (Figure 1), where an online gas chromatography-mass spectrometry/flame ionization detector system (GC-FID/MSD, TH 300B, Wuhan) is used to measure VOCs in the ambient atmosphere. The system has a sampling rate of 60 mL/min for 5 minutes per sample, with a sampling frequency of once per hour (Meng et al., 2022). The ISOP observation data undergo rigorous quality control, which can be used for evaluating simulated ISOP concentrations. It is worth noting that the ISOP observational data for the Modiesha site covers September 2017, while the Wanqingsha site has data coverage from September 7 to September 30, 2017.*

**Comment 6**: Title: "Unheralded" here in the title seems to suggest that no study before has attributed biogenic VOCs from urban greening to ozone production before, but there are many past studies that are cited in the introduction that have also concluded this. Could the authors choose a different word that better reflects the advancements and scope of their work?

**Reply:** Thanks for the good suggestion, and we have replaced the "Unheralded" with "Underappreciated" in the title.

*Underappreciated contributions of biogenic volatile organic compounds from urban greening to ozone pollution: a high-resolution modeling study.*

**Comment 7**: Line 21 – Can you expand more on what you mean by "advocated for mitigating urban atmospheric environment"? I'm not sure what this means.

**Reply:** Thanks for the good suggestion, and we have rewritten this sentence as follows.

*Urban Green Spaces (UGS), such as parks, and gardens, are widely promoted as a strategy for improving the urban atmosphere and environmental health.*

**Comment 8**: Table 3 – I believe that this is an hourly comparison between $O_3$ and $NO_2$. If so, can you add hourly to the title or table description for clarity to the reader.

**Reply:** Thanks for the good suggestion, and we have reorganized this table as follows.

*Table 2: Evaluation results of the simulated monthly mean hourly $O_3$, MDA8 $O_3$, and hourly $NO_2$ mixing ratios for each case during September 2017.*

| Pollutant | Case name | Sim (ppb) | Obs (ppb) | MB (ppb) | NMB | NME | R |
|---|---|---|---|---|---|---|---|
| Hourly $O_3$ | Gdef_N | 28.23 | 30.49 | -2.26 | -6.7% | 23.6% | 0.82 |
| | Gdef_Y | 28.67 | 30.49 | -1.82 | -5.3% | 23.6% | 0.82 |
| | Ghr_N | 28.89 | 30.49 | -1.60 | -4.8% | 22.5% | 0.83 |
| | Ghr_Y | 29.33 | 30.49 | -1.15 | -3.4% | 22.4% | 0.83 |

| | | | | | | | |
|---|---|---|---|---|---|---|---|
| MDA8 O$_3$ | Gdef_N | 60.11 | 62.27 | -2.16 | -3.47% | 21.71% | 0.84 |
| | Gdef_Y | 61.04 | 62.27 | -1.23 | -1.97% | 21.40% | 0.84 |
| | Ghr_N | 61.07 | 62.27 | -1.20 | -1.92% | 21.28% | 0.84 |
| | Ghr_Y | 62.00 | 62.27 | -0.26 | -0.42% | 21.23% | 0.84 |
| Hourly NO$_2$ | Gdef_N | 24.78 | 21.50 | 3.27 | 15.2% | 45.7% | 0.63 |
| | Gdef_Y | 24.74 | 21.50 | 3.24 | 15.0% | 45.5% | 0.63 |
| | Ghr_N | 24.35 | 21.50 | 2.84 | 13.2% | 43.8% | 0.63 |
| | Ghr_Y | 24.32 | 21.50 | 2.81 | 13.0% | 43.6% | 0.63 |

**Comment 9**: Line 272 and Table 4 – You mention MDA8 O$_3$ here in the text? Is Table 4 MDA8 O$_3$ or hourly O$_3$? If possible, calculating these statistics on hourly O$_3$ and MDA8 O$_3$ is the most useful for both Table 3 and 4. Hourly O$_3$ especially R helps understand if you have represented the diurnal cycle well. And MDA8 ozone is useful to just investigate daytime ozone when O$_3$ is highest for regulatory applications. I would recommend calculating both MDA8 O$_3$ and hourly O$_3$ for both tables here.

**Reply:** Yes, Table 4 is for MDA8 O$_3$, and we have added the evaluation of hourly O$_3$ in Table 3 and Table 4. Meanwhile, we rewritten the related text in the manuscript. Please also see our responses to the above comment.

*Additionally, various statistical metrics were used to assess the performance of hourly O$_3$, MDA8 O$_3$, and NO$_2$ concentrations from the CMAQ simulation (Emery et al. 2017). These metrics comprise the correlation coefficient (R), normalized mean bias (NMB), and normalized mean error (NME). The formulas for these metrics are listed in Table S3. As shown in Table 3, the modeling performance for all cases are reasonably, albeit with some degree of underestimation. Despite these discrepancies, the model demonstrates sufficient reliability and can be effectively used in the subsequent study. Meanwhile, the MBs of MDA8 O$_3$ across various cases indicate a substantial improvement in the CMAQ simulation when UGS-BVOC, UGS-LUCC, and their combined effects are considered. Specifically, the MB values decrease from -2.16 ppb in the Gdef_N case to -0.26 ppb in the Ghr_Y case, demonstrating that incorporating UGS-BVOC, UGS-LUCC, and their combined effects can enhance the accuracy of predicted daytime O$_3$ concentrations. In addition, we also evaluate the simulation performance for NO$_2$ in each case and the results suggest that all models have R above 0.63, and while there is some overestimation, the NMB is 15.0%, 15.2%, 13.0%, and 13.2% for Gdef_N, Gdef_Y, Ghr_N, and Ghr_Y, respectively. It should be emphasized that integrating UGS-BVOC into the modeling process can slightly improve the accuracy of NO$_2$ predictions, reducing the MB from 3.27 to 3.24 ppb, and from 2.84 to 2.81 ppb for Gdef and Ghr cases, respectively. The improvement in NO$_2$ predictions is attributed to the increased involvement of NO$_2$ in O$_3$ formation caused by the UGS-BVOC emissions, which reduces simulated NO$_2$ concentrations and narrows its bias against the observation.*

*In terms of O$_3$, the UGS-BVOC, UGS-LUCC, and their combined effects have various performances in different regions (Table 4). These results indicate that the inclusion of UGS-BVOC emissions remarkable influences MDA8 O$_3$ and hourly O$_3$ concentrations in the city center region and this effect, primarily observed when comparing the Gdef_Y with Gdef_N and Ghr_Y with Ghr_N cases, is largely due to the VOC-limited*

*areas prevalent in Guangzhou (He et al., 2024). By integrating the UGS-BVOC emissions into the models (comparing Gdef_Y and Gdef_N cases), the MBs of MDA8 $O_3$ and hourly $O_3$ in all regions, including a notable improvement in the city center region from -3.62 to -0.75 ppb and -2.86 to -1.18 ppb, respectively, is reduced. Additionally, the UGS-BVOC emissions slightly enhance R values of MDA8 $O_3$ and hourly $O_3$ in the city center and suburban regions, indicating a more accurate the daytime trend and the diurnal cycle representation, respectively. The UGS-LUCC effects, as seen when comparing Ghr_N and Gdef_N cases, also greatly improve model biases and the combined effects of both UGS-BVOC and UGS-LUCC (comparing the Ghr_Y and Gdef_N cases) substantially ameliorate model biases in the city center and suburban regions.*

*Table 4 Evaluation results of simulated monthly mean hourly $O_3$ and MDA8 $O_3$ mixing ratios in city center, suburban, and rural areas for each case during September 2017.*

| Variable | Regions | MB (ppb) | | | | R | | | |
|---|---|---|---|---|---|---|---|---|---|
| | | Gdef_N | Gdef_Y | Ghr_N | Ghr_Y | Gdef_N | Gdef_Y | Ghr_N | Ghr_Y |
| MDA8 $O_3$ | City center | −3.627 | −2.241 | −2.110 | −0.747 | 0.805 | 0.810 | 0.810 | 0.813 |
| | Suburban | −4.076 | −3.251 | −3.210 | −2.376 | 0.737 | 0.743 | 0.717 | 0.727 |
| | Rural | −5.109 | −4.757 | −4.866 | −4.528 | 0.665 | 0.655 | 0.695 | 0.690 |
| Hourly $O_3$ | City center | -2.862 | -2.292 | -2.086 | -1.520 | 0.800 | 0.802 | 0.811 | 0.812 |
| | Suburban | -3.148 | -2.803 | -2.647 | -2.295 | 0.824 | 0.825 | 0.824 | 0.826 |
| | Rural | -1.184 | -1.630 | -1.375 | -1.164 | 0.742 | 0.741 | 0.751 | 0.750 |

**Comment 10**: Line 288 – By "monoethylene" here, do you mean "monoterpene". Is this a typo?

**Reply:** Thanks for your careful check, and we have fixed this typo.

*A review of the data reveals that TERP and ISOP rank as the highest emitting species with proportions are 20.46% and 31.91% in this study, respectively, aligning with the findings of previous studies (Cao et al., 2022; Guenther et al., 2012b).*

**Comment 11**: Line 297 – See major comment. I would suggest renaming urban in figure 1 and throughout the text including in this paragraph to something else so that readers can clearly differentiate between urban landcover type and what you are classifying as an urban region.

**Reply:** Thanks for the valuable suggestion, and we have renamed the "urban" as "city center" in Figure 1 and all related text in the manuscript.

**Comment 12**: Line 381 – Can you verify this statement "high-resolution land use cover data increase the estimation of the UGS-BVOC emissions in the urban and suburban regions." This is different from the text above and Figure 2A.

**Reply:** Thanks for the carefully check, we have rewritten this sentence.

*The analysis also highlights high-resolution land use cover data increase the estimation of the UGS-BVOC emissions in the city center region.*

**Comment 13**: Throughout 3.2, and all figures and tables therein, can you be clearer in the table and figure descriptions and throughout the text that these are all evaluated over September. I assume this is the case, but it would make it easier for the reader to state this clearly in the table and figure headings and descriptions.

**Reply:** Thanks for the carefully check, we have rewritten this sentence.

**Comment 14**: Lines 397 – 400 – These are very strong statements, but it's hard to confirm that these statements are really accurate without also plotting the absolute concentrations of MDA8 ozone and how much the UGS-BVOC and UGS-LUCC contribute to the total. Your conclusions here would be more impactful, if you plotted the total MDA8 ozone and then the contribution from UGS-BVOC and UGS-LUCC and where the regulatory metric is for MDA8 ozone as well. I would suggest showing a plot of your total MDA8 as well.

**Reply:** Thanks for the nice suggestion, we have softened the statements.

*The observed increase suggests a potentially significant influence of UGS-BVOC emissions and UGS-LUCC on ozone levels, indicating that these factors may play an important role in ozone pollution research and should be carefully considered.*

**Comment 15**: Figure 6 – If this is an average over September of the differences? Can you add this to the Figure description for clarity to the reader.

**Reply:** Thanks for the nice suggestion, we have changed the title for the figure.

[Figure]

*Figure 1 The map of UGS-BVOC effects (a), LUCC effects (b), and combined effects (c) in MDA8 O₃. Each map shows the difference in average MDA8 O₃ concentrations for each case (Gdef_Y, Ghr_N, and Ghr_Y) relative to the Gdef_N case during September 2017.*

**Comment 16**: Figure 7A – Why plot your base case scenario (Gdef_N) here for MDA8 ozone. Why not your improved scenario (Ghr_Y) or at least both?

**Reply:** Thanks for the nice suggestion, we have replotted the figure using the simulation from the base case.

[Figure]

*Figure 2 The comparison during September 2017 between the average values from simulation results grids which have air quality stations produced by the Gdef_N case and the average observation values for MDA8 O₃ (A). (B) is the meteorological fields from the average values from the simulation result grids, which have the same locations as the air quality stations. (C) is the observed average values for NO₂ and CO concentrations from all air quality stations.*

**Comment 17**: Line 459 – What do you mean by "cumulative effects" here?

**Reply:** It means the combined effects of UGS-LUCC and UGS-BVOC. We have rewritten this sentence.

*Furthermore, the UGS-LUCC effect's maximal contribution to the urban MDA8 O₃ levels could escalate to 2.2 ppb in Episode 1 and 23.7 ppb in Episode 2 while the combined effects of UGS-LUCC and UGS-BVOC emissions are projected to enhance MDA8 O₃ concentrations to 4.8 ppb and 25.2 ppb for the respective episodes.*

**Comment 18**: Line 519 – I'm not sure these are what everyone would consider significant changes in ozone. Please add the percent changes in ozone in parenthesis here, so that readers can confirm your definition of significant is similar to their definition.

**Reply:** Thanks for this commend, and we have rewritten this sentence.

*The UGS-BVOC emissions have a remarkable impact on ozone concentrations, with increases ranging from 1.0-1.4 ppb (+2.3-3.2%) in the city center regions.*

**Comment 19**: Data availability – ACP journals require at least model code be made public (https://www.atmospheric-chemistry-and-physics.net/policies/data_policy.html). Please upload your WRF-Chem model code to be available online.

**Reply:** Thanks for the reminder, and we have rewritten this part.

*The WRF (Weather Research and Forecasting Model) code can be obtained from the official repository at https://github.com/wrf-model/WRF. The CMAQ (Community Multiscale Air Quality Model) code is accessible at https://github.com/USEPA/CMAQ. Model output data used for analysis and plotting, and the code used for simulations can be made available upon request (Haofan Wang, wanghf58@mail2.sysu.edu.cn).*

**Comment 20**: Line 210 need a space between Table 1 and were.

**Reply:** Thanks for the carefully check, and we have added a space between "Table 1" and "were".

*In this study, four distinct cases, as listed in Table 1 were established to investigate the impacts of UGS-LUCC, UGS-BVOC, and their combined effects on the ozone simulation.*

**Comment 21**: Line 507 – not a complete sentence.

**Reply:** Thanks for the carefully check, and we have rewritten this sentence.

*Considering the UGS-BVOC and UGS-LUCC effects can effectively mitigate the underestimation of surface ozone concentrations by regional air quality models, though other factors such as inaccuracies in emissions inventories, chemical mechanisms, and meteorological inputs may also contribute to these underestimations.*

**Comment 22**: Line 520 – typo with a comma instead of a period.

**Reply:** Thanks for the carefully check, and we have rewritten this sentence.

*The UGS-BVOC emissions have a remarkable impact on ozone concentrations, with increases ranging from 1.0-1.4 ppb (+2.3-3.2%) in the city center regions.*

---

## Author Comment (AC3)

**Response to Reviewer #2:**

We gratefully thank the editor and all reviewers for their time spent making their constructive remarks and useful suggestions, which have significantly raised the quality of the manuscript and have enabled us to improve the manuscript. Each suggested revision and comment, brought forward by the reviewers was accurately incorporated and considered. Below are the comments of the reviewers and response point by point and the revisions are indicated. We use different colored fonts to distinguish between responses to reviewers and the revised sections of the manuscript.

1. Responses to reviewers are highlighted in blue.

2. *Revised sections of the manuscript are highlighted in red.*

**Comment 1**: This manuscript describes a numerical modeling study (WRF-CMAQ-MEGAN) to investigate the impact of urban greening on air pollution. This is an important topic that has been the subject of quite a few papers recently but there is certainly more that needs to be done to adequately address this topic. The authors use a relatively high-resolution (1 km) modeling system which is appropriate for advancing our understanding on urban BVOC impacts. They compare their results with observations which is helpful for seeing if there are improvements. They then present three scenarios (landcover data resolution, BVOC emission estimates, urban green space amount) to a base case to show the impacts of each one. This is a useful study but I have some concerns about the manuscript that should be addressed before accepting this paper for publication in ACP.

**Reply:** We sincerely thank the reviewer for their thoughtful and encouraging feedback on our study. We are pleased that you recognize the importance of investigating urban greening impacts on air pollution and appreciate your acknowledgment of our approach, including the use of high-resolution modeling and the comparison with observations. Your comments motivate us to continue refining our work and addressing the specific concerns raised to ensure the manuscript meets the high standards of ACP.

**Comment 2** The introduction (Section 1) has statements that do not seem to be supported by the references given and these references do not appear to be very relevant for urban green spaces anyway. Examples include lines 63-64, 67-71, 71-73, 80-81, etc.

**Reply:** Thanks for this suggestion. We have added some relevant references.

*Deposition involves the absorption of air pollutants onto vegetative surfaces, while dispersion refers to the reduction of air pollutant concentrations through aerodynamic effects caused by vegetation (Tiwari and Kumar, 2020; N. Wang et al., 2019a).*

*These conditions are influenced by several factors, such as the specific structure of the UGS vegetation properties (e.g., height, leaf density), the site context (e.g., street canyon geometry, proximity to emission sources), and prevailing meteorological conditions (e.g., wind speed and direction) (Jin et al., 2017; Tomson et al., 2021; Yang et al., 2020).*

*For example, dense tree canopies might impede ventilation in urban street canyons, while porous vegetation barriers in open-road settings could potentially intensify roadside air pollution concentrations (Chen et al., 2021; Jin et al., 2014).*

*UGS also have a complex role in air quality due to their production of biogenic volatile organic compounds (BVOCs). For instance, in cities like Los Angeles, the UGS-BVOC emissions contribute to a quarter of the secondary organic aerosol formation on hot days (Schlaerth et al., 2023).*

*Chen, X., Wang, X., Wu, X., Guo, J., Zhou, Z., 2021. Influence of roadside vegetation barriers on air quality inside urban street canyons. Urban For. Urban Green. 63, 127219. https://doi.org/10.1016/J.UFUG.2021.127219*

*Fu, X., Liu, J., Ban-Weiss, G., Zhang, J., Huang, X., Ouyang, B., Popoola, O., Tao, S., 2017. Effects of canyon geometry on the distribution of traffic-related air pollution in a large urban area: Implications of a multi-canyon air pollution dispersion model. Atmos. Environ. 165, 111–121. https://doi.org/10.1016/J.ATMOSENV.2017.06.031*

*Jin, S., Guo, J., Wheeler, S., Kan, L., Che, S., 2014. Evaluation of impacts of trees on PM2.5 dispersion in urban streets. Atmos. Environ. 99, 277–287. https://doi.org/10.1016/J.ATMOSENV.2014.10.002*

*Schlaerth, H.L., Silva, S.J., Li, Y., 2023. Characterizing Ozone Sensitivity to Urban Greening in Los Angeles Under Current Day and Future Anthropogenic Emissions Scenarios. J. Geophys. Res. Atmospheres 128, e2023JD039199. https://doi.org/10.1029/2023JD039199*

*Tiwari, A., Kumar, P., 2020. Integrated dispersion-deposition modelling for air pollutant reduction via green infrastructure at an urban scale. Sci. Total Environ. 723, 138078. https://doi.org/10.1016/j.scitotenv.2020.138078*

*Tomson, M., Kumar, P., Barwise, Y., Perez, P., Forehead, H., French, K., Morawska, L., Watts, J., 2021. Green infrastructure for air quality improvement in street canyons. Environ. Int. 146, 106288. https://doi.org/10.1016/j.envint.2020.106288*

*Xing, Y., Brimblecombe, P., 2019. Role of vegetation in deposition and dispersion of air pollution in urban parks. Atmos. Environ. https://doi.org/10.1016/J.ATMOSENV.2018.12.027*

*Yang, H., Chen, T., Lin, Y., Buccolieri, R., Mattsson, M., Zhang, M., Hang, J., Wang, Q., 2020. Integrated impacts of tree planting and street aspect ratios on CO dispersion and personal exposure in full-scale street canyons. Build. Environ. 169, 106529. https://doi.org/10.1016/j.buildenv.2019.106529*

**Comment 3**: The comparison between the "default" low resolution run and the higher resolution data there are other differences in these runs and they do not really show what is the impact of 1) the resolution of the data used to derive landcover and 2) the resolution of the model simulations. It would be more useful to show the individual impacts of these two differences. Also, while it is reasonable to assume that 10-m landcover data is more appropriate than 1-km landcover data, there are still uncertainties associated with the 10-m data. There should be a detailed discussion of the uncertainties for each landcover input (LAI, growth form fractions, ecotypes and ecotype-specific emission factors) and how this compares with 1-km data.

**Reply:** Thanks for this comment, and we have added a section "Uncertainties and limitations" to address this comment.

*In this study, we used land use and land cover data integrated at 1-km and 10-m resolutions to define the urban boundary and characterize the spatial distribution of UGS in Guangzhou. Additionally, we incorporate high-resolution LAI data, obtained through machine learning, as input for the MEGAN model. Using the WRF-CMAQ model, we quantify the effects of UGS-BVOC, UGS-LUCC, and their combined impacts on ozone concentrations in Guangzhou. However, some uncertainties and limitations remain.*

*First, the 10-m resolution land use and land cover data still cannot fully capture the spatial pattern of UGS in Guangzhou. As shown in Figure S2, although UGS in Guangzhou is primarily composed of EBTs, most of these EBTs are distributed along urban edges. This may result from distortions in the definition of urban extent, such as misclassifying mixed urban-vegetation grids as urban grids, caused by the coarse resolution of the 1-km land use and land cover data. The fuzzy definition of urban boundaries could lead to non-UGS areas being misclassified as UGS, potentially resulting in an overestimation of UGS-BVOC emissions.*

*Second, due to resolution limitations, only larger patches of grassland, cropland, and woodland are recognized as UGS, while smaller UGS vegetation, such as street trees, often goes undetected at a 10-m resolution. This omission can lead to an underestimation of the UGS-BVOC emissions.*

*Third, the 10-m and 1-km resolution land use and land cover data, along with the growth forms and ecotype data, use simplified categorizations for grids, which cannot fully capture the diversity of vegetation species within UGS. Since different vegetation species have varying emission factors, this simplification introduces some errors. Similarly, the oversimplified classification of land grids limits this study's ability to provide specific planning strategies for UGS at the species level. Nevertheless, it can highlight the importance of considering UGS-BVOC and UGS-LUCC in air pollution prevention and control policies.*

**Comment 4**: The comparison with ambient isoprene observations is a valuable addition to this study but it should be extended by discussing the issues with this limited comparison including how these concentrations are influenced by model emissions, dilution, chemical losses, etc which have major uncertainties. How do uncertainties compare with the differences in the results for the different scenarios? Also, include a description of the BVOC data used to evaluate the model and an assessment of the quality of the data.

**Reply:** Thanks for this comment, and we have added the description of the isoprene measurement. The validation results are constrained by the limited ISOP observations from only 2 sites. Additionally, the

estimation of BVOC emissions by the MEGAN model is inherently uncertain, which may further contribute to validation bias.

*For the isoprene (ISOP) evaluation, we use observation data from the Modiesha (23.11°N, 113.33°E) and Wanqingsha (22.71°N, 113.55°E) sites (Figure 1), where an online gas chromatography-mass spectrometry/flame ionization detector system (GC-FID/MSD, TH 300B, Wuhan) is used to measure VOCs in the ambient atmosphere. The system has a sampling rate of 60 mL/min for 5 minutes per sample, with a sampling frequency of once per hour (Meng et al., 2022). The ISOP observation data undergo rigorous quality control, which can be used for evaluating simulated ISOP concentrations. It is worth noting that the ISOP observational data for the Modiesha site covers September 2017, while the Wanqingsha site has data coverage from September 7 to September 30, 2017.*

**Comment 5**: A major missing component is a thorough discussion on what these findings can tell us about urban BVOC and air pollution- more than ozone will increase if there is sufficient NOx which is already well known. This needs to include a discussion of NOx as well as other complexities. Make it clear whether these findings are specific for Guangzhou or are also relevant for other cities. Discuss limitations such as not having representative BVOC emission factor data for these landscapes (that requires tree species composition data and accurate tree species-specific emission factors). Also, what are the uncertainties associated with growth form estimates. If the UGS is mostly grass then that will have a much lower emission than trees.

**Reply:** Thanks for this comment, and we have added related discussion to Section 4.

*Finally, Guangzhou, the study area, is a highly urbanized Chinese metropolis with a VOC-limited region (Gong et al., 2018; Kai et al., 2011; Liu et al., 2021). As a result, even a relatively small amount of VOC emissions, such as those from UGS-BVOC, can significantly impact ozone concentrations. Therefore, policymakers in Guangzhou should prioritize addressing the role of UGS-BVOC emissions in air pollution prevention and control. In other cities, particularly those with advanced urban development, high $NO_x$ emissions—often resulting from factors like high motor vehicle ownership—can lead to VOC-limited conditions. In such areas, it is equally important to emphasize the role of UGS-BVOC emissions in ozone pollution. In contrast, cities with lower $NO_x$ emissions identified as $NO_x$-limited regions may experience minimal impact from UGS-BVOC emissions on ozone concentrations.*

*Gong, J., Hu, Z., Chen, W., Liu, Y., Wang, J., 2018. Urban expansion dynamics and modes in metropolitan Guangzhou, China. Land Use Policy 72, 100–109. https://doi.org/10.1016/J.LANDUSEPOL.2017.12.025*
*Kai, Z., Wen-jie, Z., Zhi-fang, W., Wei, C., Shao-lin, P., 2011. The influence of urbanization on atmospheric environmental quality in Guangzhou, China. 2011 Int. Conf. Electr. Technol. Civ. Eng. ICETCE 3667–3670. https://doi.org/10.1109/ICETCE.2011.5776281*

*Liu, Z., Doherty, R., Wild, O., Hollaway, M., O'Connor, F., 2021. Contrasting chemical environments in summertime for atmospheric ozone across major Chinese industrial regions: the effectiveness of emission control strategies. Atmospheric Chem. Phys. https://doi.org/10.5194/ACP-21-10689-2021*

**Comment 6**: The authors discuss the impacts of LUCC on temperature and solar radiation- the temperature "heat island" is well known but the impact on solar radiation is less clear. What are the processes in the model and are they realistic? By "urban region receives less solar radiation than other regions likely due to the shading effect of urban canopies" do you mean that you are using the ground surface temperature and light to drive BVOC emissions? The ground surface values are not the light and temperature that you should be using to drive the BVOC emissions. It should be the canopy light and temperature. Reassess whether the impacts you are seeing are influenced by the model is using the wrong light and temperature.

**Reply:** Thanks for this nice question. We don't mean that we used the ground surface temperature and light to drive BVOC emissions. MEGANv3.1 use the 2-m temperature variable from the WRF model to calculate the BVOC emissions, and the 2-m temperature in WRF can be affected by the shading effect.

**Comment 7**: Ozone responds to temperature for multiple reasons. Quantify the impact of BVOC emission response to temperature relative to these other reasons.

**Reply:** This is a valuable comment. However, we think this question beyond the scope of this study. Another study by our group (Li et al., 2024) demonstrated the responses of $O_3$ to temperature through multiple mechanisms, including changes in chemical reaction rates, BVOCs emissions, soil $NO_x$ emissions, dry deposition, PAN decomposition, and etc. The major one among all mechanisms varies in regions. Please refer to more information in this paper.

Li, S., Lu, X., and Wang, H.: Anthropogenic emission controls reduce summertime ozone-temperature sensitivity in the United States, EGUsphere [preprint], https://doi.org/10.5194/egusphere-2024-1889, 2024.

**Comment 8**: Summarize the data shown in section 3.3 in a table that provides an overview of these results.

**Reply:** Thanks for this suggestion, and we have added a table to summarize these results.

*Table 6 presents the overall results for the impacts of UGS-LUCC and UGS-BVOC on MDA8 $O_3$ concentrations. The effects show slight variations across different regions during September, while the effects during the two episodes exhibit more significant changes. In the city center region, which shows the largest changes, the UGS-BVOC effect shows increases by +1.6 ppb in Episode 1 and +5.9 ppb in Episode 2, indicating that the UGS-BVOC effects influence MDA8 $O_3$ concentrations in the city center during ozone*

*episodes, while their impact is minimal in suburban and rural regions. These results highlight the important effects of UGS-LUCC and UGS-BVOC in urban areas, especially during O₃ pollution periods.*

*Table 1 Summary of Average MDA8 O$_3$ Concentrations (ppb) for Various Effects during September 2017.*

| Regions | Periods | UGS-BVOC effect | UGS-LUCC effect | Combined effect |
|---|---|---|---|---|
| City center | Monthly | +0.4 | 0.0 | 0.4 |
| | Episode 1 | +1.6 | +1.6 | +3.2 |
| | Episode 2 | +2.9 | +5.9 | +8.9 |
| Suburban | Monthly | +0.4 | 0.0 | 0.4 |
| | Episode 1 | +1.5 | +0.9 | +2.0 |
| | Episode 2 | +1.6 | +1.7 | +3.4 |
| Rural | Monthly | +0.4 | 0.0 | 0.4 |
| | Episode 1 | +0.5 | +0.6 | +1.1 |
| | Episode 2 | +0.5 | 0.0 | +0.5 |

**Comment 9**: The manuscript would benefit from editing for English language usage.

**Reply:** Thank you for your suggestion. We have carefully reviewed and improved the English language throughout the manuscript to enhance its clarity and readability.

**Comment 10**: Line 1: "Unheralded" is not justified for the title and should be deleted.

**Reply:** We have reworded this title.

*Underappreciated contributions of biogenic volatile organic compounds from urban greening to ozone pollution: a high-resolution modeling study*

**Comment 11**: 27: In addition to the 666 Gg BVOC value for Guangzhou, include the emission per area and also how BVOC compare to AVOC.

**Reply:** Thanks for this suggestion. We have added some information about this in the abstract and main text.

*Abstract*:

*Our findings indicate that the UGS-BVOC emissions in Guangzhou amounted to 666 Gg (~90 Mg/km$^2$), with isoprene (ISOP) and monoterpene (TERP) contributing remarkably to the total UGS-BVOC emissions. In comparison to anthropogenic VOC (AVOC) and BVOC emissions, UGS-BVOC emissions account for approximately 33.45% in the city center region.*

*Section 3.2:*

*Furthermore, Table 5 reveals that in September, the UGS-BVOC emissions in Guangzhou amounted to 666 Gg (~90 Mg/km$^2$), with ISOP and TERP contributing remarkably to the total UGS-BVOC emissions. In comparison to anthropogenic VOC (AVOC) and BVOC emissions, UGS-BVOC emissions account for approximately 33.45% in the city center region.*

**Comment 12**: 28: What does it mean 30% of urban ISOP? Is the other 70% anthropogenic isoprene or is it other biogenic isoprene. Along these lines, is UGS BVOC all BVOC? Or are there BVOC from other vegetation such as street trees which are generally not thought of as green space vegetation.

**Reply:** Thanks for this question and this description is misleading. We have removed this sentence.

**Comment 13**: 36: how does this study highlight the need for selecting low-emission vegetation and refining vegetation classification? I see little here that provides specific guidance to air quality managers.

**Reply:** Thanks for this question. We have removed this description.

**Comment 14**: 63-64: Clarify the point being made regarding the importance of dispersion over deposition. What do the references say that actually support this?

**Reply:** We have added this reference, and we also briefly describe the method adopted in this paper.

*Notably, Ramanathan et al. (2001) reported that dispersion effects are significantly more impactful than deposition, exceeding it by an order of magnitude via a radiative forcing modeling method.*

**Comment 15**: Line 126 to 129: This is not the default LAI for MEGAN3.1. The LAI data referred to here is probably the MEGAN2.1 LAI data. It can be used with MEGAN3.1 but there is no default MEGAN3.1 LAI data. It is expected that users generate their own LAI data for any MEGAN 3.1 simulation.

**Reply:** Thanks for the valuable suggestion, and we have rewritten this sentence.

*The default LAI dataset to drive the MEGANv2.1 model which can used for MEGANv3.1 is derived from the enhanced Moderate Resolution Imaging Spectroradiometer (MODIS) /MOD15A2H in 2003 with 1 km spatial resolution (Myneni et al., 2015).*

**Comment 16**: Line 155-164: These equations and text describe MEGAN2.1, not MEGAN3.1.

**Reply:** Thanks for the valuable suggestion, and we have changed this part.

*The emission activity factor (γ) considers emission responses to changes in environmental and phenological conditions. Compare with earlier versions, γ in MEGANv3.1 adds quantifications for responses to high and low temperature, high wind speed, and air pollution ($O_3$).*

$$\gamma = LAI \times \gamma_{TP} \times \gamma_{LA} \times \gamma_{SM} \times \gamma_{HT} \times \gamma_{LT} \times \gamma_{HW} \times \gamma_{CO_2} \times \gamma_{BD} \times \gamma_{O_3} \qquad (Eq.\ 2)$$

*In this equation, the activity factor denotes the emission response to canopy temperature/light($\gamma_{TP}$), leaf age ($\gamma_{LA}$), soil moisture ($\gamma_{SM}$), high temperature ($\gamma_{HT}$), low temperature ($\gamma_{LT}$), high wind speed ($\gamma_{HW}$), ambient $CO_2$ concentration ($\gamma_{CO_2}$), bidirectional exchange ($\gamma_{BD}$), $O_3$ exposure ($\gamma_{O_3}$), and Leaf Area Index (LAI). In this study, $\gamma_{CO_2}$ was not considered in the BVOC emission estimation. The MEGANv3.1 approach can calculate the emissions at each canopy level as the product of the emission factor and emission activity at each level.*

**Comment 17**: Line 310-311: How do you define "natural area" vs "UGS".

**Reply:** Thanks for this suggestion, and we have changed the "natural area" to "non-UGS area".

*Figure 2B offers a clear depiction of the proportion of UGS-BVOC emissions relative to non-UGS area BVOC emissions in each region of Guangzhou City, which presents that the UGS-BVOC emissions in the city center region constitute 57.34% of the total BVOC emissions in this region because of the larger urban proportions in the city center region (Figure 5), while the UGS-BVOC emission proportion in suburban and rural are 19.44% and 1.86% respectively.*

**Comment 18**: Line 398- this is not clear. Please reword this sentence

**Reply:** Thanks for this suggestion. We have rewritten this sentence.

*This finding underscores the essential role that integrated urban planning and environmental management play in controlling ozone pollution within metropolitan regions. By considering UGS-BVOC emissions in air quality models and management plans, managers can make more informed decisions to mitigate ozone levels and improve regional air quality.*

**Comment 19**: Line 399-400: If there is a "revelation" here then you should make it clear how managers can use this and what they should be doing.

**Reply:** Thanks for this suggestion. We have rewritten this sentence.

*This finding underscores the essential role that integrated urban planning and environmental management play in controlling ozone pollution within metropolitan regions. By considering UGS-BVOC emissions in air quality models and management plans, managers can make more informed decisions to mitigate ozone levels and improve regional air quality.*

**Comment 20**: 486: replace "was" with "is" unless you are referring to a specific episode (in which case make that clear)

**Reply:** Thanks for this nice suggestion.

*The rapid urbanization process is accompanied by a higher frequency of ozone episodes.*

**Comment 21**: 488-490: This meaning of this sentence is not clear.

**Reply:** Thanks for this nice suggestion. We have rewritten this sentence.

*Guangzhou, located in southern China and known as a pioneer city in reform and opening-up policies, has experienced rapid urbanization over the past thirty years, leading to increased challenges with ozone pollution.*

**Comment 22**: 27 and 497: should note that isoprene and terpenes are only half of the total. "primary" could lead readers to think they are almost all of it.

**Reply:** Thanks for this nice suggestion.

*In September 2017, the UGS-BVOC emissions in Guangzhou totaled 666 Gg, with ISOP and TERP as the major species, emitting 213 and 136 Gg, respectively.*

**Comment 23**: 498: are there UGS outside of urban areas? I would expect that they should all be urban.

**Reply:** Yes, there are UGS outside of the city center area (i.e., suburban, and rural region). See Fig. S1.

[Figure]

*Figure S1 The processed land cover dataset. (a) is the MODIS land cover without UGS and (b) has characterized the UGS base on MODIS land cover.*

**Comment 24**: 505-506: Are you saying that managers should not plant EBTs? But some EBTs have low emissions- lower than other vegetation. They should not be all grouped together.

**Comment 25**: 507-508- should also point out that there are other possibilities for ozone underestimation.

**Reply:** Thanks for this nice suggestion. We have rewritten this sentence.

*Considering the UGS-BVOC and UGS-LUCC effects can effectively mitigate the underestimation of surface ozone concentrations by regional air quality models, though other factors such as inaccuracies in emissions inventories, chemical mechanisms, and meteorological inputs may also contribute to these underestimations.*

---

## Author Response (AR2)

**Response to Reviewer #1:**

We gratefully thank the editor and all reviewers for their time spent making their constructive remarks and useful suggestions, which have significantly raised the quality of the manuscript and have enabled us to improve the manuscript. Each suggested revision and comment, brought forward by the reviewers was accurately incorporated and considered. Below are the comments of the reviewers and response point by point and the revisions are indicated. We use different colored fonts to distinguish between responses to reviewers and the revised sections of the manuscript.

1. Responses to reviewers are highlighted in blue.

2. *Revised sections of the manuscript are highlighted in red.*

**Comment 1**: Thank you for the careful updates and consideration of the review comments. I really appreciate the updated comparison to the observations of isoprene especially moving to consistent time periods between the model and observations. This update gives the reader higher confidence in the model improvements.

**Reply:** We sincerely thank the reviewer for their positive feedback and for recognizing the updates made to the manuscript.

**Comment 2**: There are still several issues described below to fix too. Please be very careful when describing the statistics in Tables 2, 3, and 4. I noticed several inconsistencies. Also please include a plot for the isoprene comparisons to observations. The monthly average statistics while useful are not sufficient to show your model biases for this important variable considering that your entire study is on BVOC emission updates:

**Reply:** Thank you for your valuable feedback. We have carefully reviewed and addressed the inconsistencies in the statistics presented in Tables 2, 3, and 4. Additionally, we have included a plot comparing isoprene (ISOP) concentrations to observations, as per your suggestion. We agree that the monthly average statistics alone may not fully capture the model biases for this key variable, and the added plot will provide a clearer picture of the model's performance regarding BVOC emissions.

**Comment 3**: Line 33 – Please add "mean biases" instead of just "biases" to be more specific here. Please, double check the rounding and negative or positive sign of these numbers and be more specific here on where they are from? Looking at the text, I believe these numbers are taken from both Table 3 (monthly average for all sites) for $NO_2$ and Table 4 (monthly mean for just the city center but not considering the negative sign) for MDA8 $O_3$. Please, be more specific here as the reader would assume that $NO_2$ and $O_3$ here are for the same region, hourly values, and both have positive biases, but they are for different regions, metrics, and different signs for the biases?

**Reply:** Thank you for your valuable comment. We have clarified the description of the biases in the text to ensure better specificity. The updated sentence now reads:

*The study shows improvements in simulation mean biases for MDA8 $O_3$, from -3.63 to -0.75 ppb in the city center region.*

Additionally, we have reviewed and verified the rounding and sign of the numbers to ensure accuracy. As per your suggestion, we have also made it clear that the biases are from different regions and metrics.

**Comment 4**: Line 263 – Please consider re-wording this statement. I'm not sure what you mean: "BVOCs are the major sources of ISOP and monoterpene (TERP)".

**Reply:** We have reworded this statement as follows:

*ISOP and monoterpene (TERP) are the major species of BVOC emission, making their concentration assessment a feasible and convincing method for indirectly validating the accuracy of BVOC emission estimates.*

**Comment 5**: Table 2 - It would be particularly useful for the reader to include a plot of the observation to model comparisons for isoprene and not just the monthly averaged statistics shown in Table 2. This could be a simple diurnal plot of the month of September like Figure S5a or a scatter or box and whisker plot. The R2 values indicate correlation is only okay, so having a plot to show this variability or spread in the obs and model is very useful rather than just looking at averages. Also considering the authors focus in on a couple of high ozone episodes understanding how the model compares with observations for isoprene on those high ozone days would also be quite useful. I highly recommend adding a plot like this to your paper even if just in the supplement.

**Reply:** Thank you for the helpful suggestion. We have now added a plot (Figure S4) comparing observations to the model for isoprene, as recommended.

[Figure]

*Figure S4 The hourly variations of observed and simulated ISOP concentrations for different cases at Modiesha (a) and Wanqingsha (b) sites during September 2017.*

*In Section 3.1:*

*Additionally, all cases successfully capture the hourly ISOP concentrations when compared to observations at both the Modiesha and Wanqingsha sites (Figure S4).*

**Comment 6**: Line 294 – typo reasonably should be reasonable.

**Reply:** Thanks for this suggestion. We have changed "reasonably" to "reasonable".

*As shown in Table 3, the modeling performance for all cases are reasonable, albeit with some degree of underestimation.*

**Comment 7**: Line 298 – Add what variable and metric you are comparing here. Assuming MDA8 ozone.

**Reply:** Thanks for the nice suggestion. We have reworded this sentence as follows:

*Specifically, the MB values of MDA8 $O_3$ decrease from -2.16 ppb in the Gdef_N case to -0.26 ppb in the Ghr_Y case, demonstrating that incorporating UGS-BVOC, UGS-LUCC, and their combined effects can enhance the accuracy of predicted daytime $O_3$ concentrations.*

**Comment 8**: Line 305 - Can you rephrase this sentence? I'm not sure what you mean. "The improvement in NO2 predictions is attributed to the increased involvement of $NO_2$ in $O_3$ formation caused by the UGS-BVOC emissions, which reduces simulated $NO_2$ concentrations and narrows its bias against the observation.

**Reply:** Thanks for the valuable suggestion. We have rephrased this sentence as follows:

*The improvement in $NO_2$ predictions is attributed to the inclusion of UGS-BVOC emissions in the CMAQ model, which enhances $NO_2$ involvement in $O_3$ formation. This process leads to lower simulated $NO_2$ concentrations, reducing the MBs compared to observations.*

**Comment 9**: Line 314 - typo remarkable should be remarkably

**Reply:** Thanks for the careful checking. As suggested by another referee We have changed "remarkable" to "significantly" here.

*These results indicate that the inclusion of UGS-BVOC emissions significantly influences MDA8 $O_3$ and hourly $O_3$ concentrations in the city center region and this effect, primarily ...*

**Comment 10**: Table 4 and line 318 - 319 – The table says -3.627 and you say -3.62 in the text. How are you rounding this? Please be consistent. Also be consistent with this negative number in the abstract and conclusion as specified above and below. I do not see -1.18 ppb in the table for city center hourly ozone, please double check this number? On line 318, please double check which cases these numbers are comparing against. The way you have written this sentence, I would assume you are comparing Gdef_Y and Gdef_N cases, but that's not where the numbers are coming from?

**Reply:** We sincerely thank the reviewer for pointing out this inconsistency. In the original manuscript, the value in the text (-3.62) was rounded differently from the value in Table 4 (-3.627). To ensure consistency and accuracy, we have revised the text to reflect the value -3.63, which is the correctly rounded value of -3.627 to two decimal places.

Upon careful review, we identified an error in the originally reported value of -1.18 ppb for city center hourly ozone. The correct value, as shown in the table, is -1.52 ppb. We have updated the text to reflect this correction.

We sincerely apologize for the oversight and appreciate the reviewer's diligence in helping us improve the accuracy of our manuscript.

The reviewer is correct that the original sentence incorrectly implied a comparison between the Gdef_Y and Gdef_N cases. In fact, the numbers are derived from a comparison between the Ghr_Y and Gdef_N cases, which accounts for the integration of both UGS-BVOC emissions and UGS-LUCC. We have revised the text on line 318 to clarify this comparison and ensure accuracy. The updated sentence now reads:

*By integrating the UGS-BVOC emissions and UGS-LUCC into the models (comparing Ghr_Y and Gdef_N cases), the MBs of MDA8 $O_3$ and hourly $O_3$ in all regions, including a notable improvement in the city center region from -3.63 to -0.75 ppb and -2.86 to -1.52 ppb, respectively, is reduced.*

**Comment 11**: Line 340 – How are you rounding these numbers? Please, make sure they are just rounded versions of what is in the table or use the exact same numbers.

**Reply:** We thank the reviewer for pointing out this inconsistency. To address this, we have revised the text on line 340 to ensure that the rounded numbers in the text align with the values in the table. Specifically, we have applied consistent rounding rules (e.g., rounding to two decimal places) to the numbers in the text, ensuring they are accurate representations of the values in the table. This change improves clarity and consistency throughout the manuscript. We appreciate the reviewer's attention to detail, which has helped us enhance the precision of our presentation.

*Regionally, the suburban region registered the highest UGS-BVOC emissions in Guangzhou, peaking at 367 Gg. This is followed closely by the rural and city center regions, recording emissions of 174 Gg and 126 Gg, respectively.*

**Comment 12**: Line 545 – Typo make sure to translate the Chinese characters here for final version.

**Reply:** Thanks for the careful checking. We have deleted these Chinese characters. The revised version here is as follows:

*Table 6 presents the overall results for the impacts of UGS-LUCC and UGS-BVOC on MDA8 $O_3$ concentrations.*

**Comment 13**: Line 614 – What do you mean by "baseline concentration of 0.35 ppb"? The observations are 0.34 in Table 2? But maybe you are referring to something else. Please double check and replace baseline with something more descriptive.

**Reply:** Thanks for this nice suggestion. We have rewritten this sentence as follows:

*For instance, incorporating UGS-BVOC emissions results in an increase in ISOP concentration from 0.29 ppb to 0.35 ppb and from 0.23 ppb to 0.29 ppb under different land use cases (Gdef and Ghr), compared to a observed concentration of 0.34 ppb.*

**Comment 14**: Line 618 – Should 3.62 here also be negative? Also see above should this be rounded differently to compare with the value in Table 4. Also, if this is MDA8 ozone, please add this here to be clearer.

**Reply:** Thanks for this valuable suggestion. We have rewritten this sentence as follows:

*Secondly, this precise estimation of BVOCs and the consideration of UGS-LUCC has notably shifted the MB of MDA8 $O_3$ simulations from -3.63 ppb to -0.75 ppb in the city center region.*

**Comment 15**: Figure S5a has ozone still in ug/m3 instead of ppb, which is hard for the reader to compare with Figure 7a, which is in ppb. It would be useful if these plots all used consistent units.

**Reply:** *We thank the reviewer for pointing out this inconsistency in the units used in Figure S5a. To ensure clarity and facilitate comparison with Figure 7a, we have updated Figure S5a to use consistent units, converting ozone concentrations from µg/m³ to ppb. This change aligns with the units used in Figure 7a and improves the readability and comparability of the results. We appreciate the reviewer's suggestion, which has enhanced the consistency of our presentation.*

[Figure]

*Figure S5 The comparison between the average hourly O3 values from simulation and observation (A). And the relative grid average values of solar radiation and PBLH (B).*

**Response to Reviewer #2:**

We gratefully thank the editor and all reviewers for their time spent making their constructive remarks and useful suggestions, which have significantly raised the quality of the manuscript and have enabled us to improve the manuscript. Each suggested revision and comment, brought forward by the reviewers was accurately incorporated and considered. Below are the comments of the reviewers and response point by point and the revisions are indicated. We use different colored fonts to distinguish between responses to reviewers and the revised sections of the manuscript.

1. Responses to reviewers are highlighted in blue.

2. *Revised sections of the manuscript are highlighted in red.*

**Comment 6**: The authors discuss the impacts of LUCC on temperature and solar radiation- the temperature "heat island" is well known but the impact on solar radiation is less clear. What are the processes in the model and are they realistic? By "urban region receives less solar radiation than other regions likely due to the shading effect of urban canopies" do you mean that you are using the ground surface temperature and light to drive BVOC emissions? The ground surface values are not the light and temperature that you should be using to drive the BVOC emissions. It should be the canopy light and temperature. Reassess whether the impacts you are seeing are influenced by the model is using the wrong light and temperature.

**Reply:** Thanks for this nice question. We don't mean that we used the ground surface temperature and light to drive BVOC emissions. MEGANv3.1 use the 2-m temperature variable from the WRF model to calculate the BVOC emissions, and the 2-m temperature in WRF can be affected by the shading effect.

**RC's reply:** Please address this with a sentence in the manuscript as other readers will also likely be confused.

**Reply:** We have added a sentence to Section 2.2.

*It is worth noting that MEGANv3.1 uses the 2-m temperature variable from the WRF model to calculate BVOC emissions. Meanwhile, The MEGANv3.1 approach can calculate the emissions at each canopy level as the product of the emission factor and emission activity at each level.*

**Comment 7**: Ozone responds to temperature for multiple reasons. Quantify the impact of BVOC emission response to temperature relative to these other reasons.

**Reply:** This is a valuable comment. However, we think this question beyond the scope of this study. Another study by our group (Li et al., 2024) demonstrated the responses of $O_3$ to temperature through multiple mechanisms, including changes in chemical reaction rates, BVOCs emissions, soil $NO_x$ emissions, dry

deposition, PAN decomposition, and etc. The major one among all mechanisms varies in regions. Please refer to more information in this paper.

Li, S., Lu, X., and Wang, H.: Anthropogenic emission controls reduce summertime ozone-temperature sensitivity in the United States, EGUsphere [preprint], https://doi.org/10.5194/egusphere-2024-1889, 2024.

**RC's reply:** You can refer to the Li et al. paper, rather than making these calculations, but you should still be able to make some statement in the manuscript regarding whether the BVOC is a major component of the temperature dependent VOC or not and point out that there are other components.

**Reply:** We have added some statement to Section 3.2.

*Temperature-dependent BVOC emissions are among the well-known key temperature-dependent mechanisms influencing ozone levels, alongside other processes including changes in chemical reaction rates, soil $NO_x$ emissions, dry deposition, and PAN decomposition, as demonstrated in Li et al. (2024).*

*Li, S., Lu, X., and Wang, H.: Anthropogenic emission controls reduce summertime ozone-temperature sensitivity in the United States, EGUsphere [preprint], https://doi.org/10.5194/egusphere-2024-1889, 2024.*

**One final comment:** The use of "remarkable" on lines 314 and 623 are not warranted. The original use of "significant" is OK or just don't have an adjective there. The use of "remarkable" on line 625 is OK and distinguishes this point from the other two.

**Reply:** We have addressed these suggestions.

*Line 314:*

*These results indicate that the inclusion of UGS-BVOC emissions significantly influences MDA8 $O_3$ and hourly $O_3$ concentrations in the city center region and this effect, primarily…*

*Line 623 and 625:*

*The UGS-BVOC emissions have a significant impact on ozone concentrations, with increases ranging from 1.0-1.4 ppb (+2.3-3.2%) in the city center regions. However, when considering the combined UGS-LUCC and UGS-BVOC effects, the impact on MDA8 $O_3$ concentrations becomes remarkable, …*

---

## Author Response (AR3)

**Response to Editor:**

We gratefully thank the editor for your time spent making the constructive remarks and useful suggestions, which have significantly raised the quality of the manuscript and have enabled us to improve the manuscript. Each suggested revision and comment, brought forward by you was accurately incorporated and considered. Below are the comments of the editor and response point by point and the revisions are indicated. We use different colored fonts to distinguish between responses to reviewers and the revised sections of the manuscript.

1. Responses to reviewers are highlighted in blue.

2. *Revised sections of the manuscript are highlighted in red.*

**Comment 1**: many thanks for your revision of the manuscript as a response to the last referee comments.

I read the manuscript and have numerous small comments (see below), I would like you to address before I accept the manuscript for publication in ACP.

**Reply:** We sincerely appreciate your time and effort in reviewing our revised manuscript. Thank you for your constructive feedback and for considering our work for publication in ACP. We have carefully addressed all the comments and made the necessary revisions to further improve the manuscript. Below, we provide detailed responses to each point.

**Comment 2**: - Why not using 'Urban green spaces' in the title? E.g., Underappreciated contributions of biogenic volatile organic compounds from urban green spaces to ozone pollution

- Note that titles that highlight the method are less preferred according to the journal guidelines

https://www.atmospheric-chemistry-and-physics.net/policies/guidelines_for_authors.html

**Reply:** Thank you for your suggestion. We have revised the title to use "urban green spaces" instead of "urban greening," as recommended. Additionally, we have removed the ":a high-resolution modelling study" in accordance with the journal's guidelines. The revised title now reads:

*Underappreciated contributions of biogenic volatile organic compounds from urban green spaces to ozone pollution*

**Comment 3**: - Abstract: It is too long: Please shorten it so it adheres to the journal guidelines (250 words)

**Reply:** Thank you for your suggestion. We have revised and shortened the abstract to ensure it adheres to the journal's 250-word limit. The updated abstract now reads as follows:

*Urban Green Spaces (UGS), such as parks, and gardens, are widely promoted as a strategy for improving the urban atmospheric environment. However, this study reveals that it can exacerbate urban ozone ($O_3$) levels under certain conditions, as demonstrated by a September 2017 study in Guangzhou, China. Using the Weather Research and Forecasting Model with the Model of Emissions of Gases and Aerosols from Nature (WRF-MEGAN) and the Community Multiscale Air Quality (CMAQ) model, we assessed the impact of UGS-related biogenic volatile organic compound (BVOC) emissions on urban $O_3$. Our findings indicate that the UGS-BVOC emissions in Guangzhou amounted to 666 Gg (~90 Mg/km$^2$), with isoprene (ISOP) and monoterpene (TERP) contributing remarkably to the total UGS-BVOC emissions. Compared to anthropogenic VOC (AVOC) and BVOC emissions, UGS-BVOC emissions account for ~33.45% in the city center, and their inclusion in the model reduces ISOP underestimation. The study shows improved simulation mean biases for MDA8 $O_3$, from -3.63 to -0.75 ppb in the city center. Integrating UGS-BVOC and UGS-LUCC (Land Use Cover Change) enhances surface monthly mean $O_3$ by 1.7–3.7 ppb (+3.8–8.5%) and adds up to 8.9 ppb (+10.0%) to MDA8 $O_3$ during pollution episodes. UGS-BVOC emissions alone increase monthly mean $O_3$ by 1.0–1.4 ppb (+2.3–3.2%) in urban areas and contribute up to 2.9 ppb (+3.3%) to MDA8 O3 during pollution episodes. These impacts can extend to surrounding suburban and rural areas through regional transport, highlighting the need to accurately account for UGS-BVOC emissions to better manage air quality.*

**Comment 4**: l. 178 – 181: It is worth noting that MEGANv3.1 uses the 2-m temperature variable from the WRF model to calculate BVOC emissions. Meanwhile, The MEGANv3.1 approach can calculate the emissions at each canopy level as the product of the emission factor and emission activity at each level.

Not clear what you mean by 'meanwhile'. Should it be 'Therefore'?

**Reply:** We have removed this sentence. The follows are the revised version:

*It is worth noting that MEGANv3.1 uses the 2-m temperature variable from the WRF model to calculate BVOC emissions.*

**Comment 5**: l. 186: Here 'Meanwhile' seems redundant.

**Reply:** We have removed the "Meanwhile" here.

*The growth form datasets in MEGANv3.1 contain considerations of evergreen broadleaf forests, grasslands, and crops, which cover all types of UGS in Guangzhou city (Figure S1).*

**Comment 6**: Table 1: Please add sufficient information to the table caption that the table can be understood independently.

**Reply:** Thanks for this suggestion. We have revised this table caption.

*Table 1 Case configurations. The default land cover (LC) datasets are derived from MODIS/MCD12Q1, while the high-resolution LC datasets use MODIS/MCD12Q1 for natural areas and the 10-m datasets from Geographic Remote Sensing Ecological Network Platform for urban areas. N-LAI (None-urban Leaf Area Index) indicates that the model uses LAI data without urban LAI, whereas T-LAI (Total LAI) includes urban LAI. The "Description" column explains the purpose of each case.*

| Name | LC dataset | LAI dataset | Description |
|------|-----------|-------------|-------------|
| Gdef_N | Default data | N-LAI | Base |
| Gdef_Y | Default data | T-LAI | UGS-BVOC effects |
| Ghr_N | High-resolution data | N-LAI | UGS-LUCC effects |
| Ghr_Y | High-resolution data | T-LAI | combined effects |

**Comment 7**: l. 268: What do you mean by "delineated within this study"? I suggest removing it.

**Reply:** Thanks for the carefully suggestion. We have removed this.

*Table 2 presents the mean concentrations of ISOP derived from various cases juxtaposed with the observed average concentrations.*

**Comment 8**: l. 306: 'Modeling process' could be simply replaced by 'model'

**Reply:** Thanks for the valuable suggestion. We have rephrased this sentence as follows:

*It should be emphasized that integrating UGS-BVOC into the model can slightly improve the accuracy of $NO_2$ predictions, reducing the MB from 3.27 to 3.24 ppb, and from 2.84 to 2.81 ppb for Gdef and Ghr cases, respectively.*

**Comment 9**: l. 332: 'the primary' can be removed

**Reply:** Thanks for the carefully suggestion. We have removed this.

*Given that the variances in the UGS-BVOC emissions due to different land use covers are relatively minor, Table 5 presents emissions driven by the default land use cover.*

**Comment 10**: l. 334: 'highest emitting species' should be either 'highest emissions' or 'species with highest emissions'.

'Emitting species' does not make sense as it would imply that the species are emitting something.

**Reply:** Thanks for the valuable suggestion. We have rephrased this sentence as follows:

*A review of the data reveals that TERP and ISOP rank as the highest emissions with proportions are 20.46% and 31.91% in this study, respectively, ...*

**Comment 11**: Figure 2: Please improve the figure caption to clearly describe what is shown in each panel.

**Reply:** Thanks for the valuable suggestion. We have revised the caption of Figure 2 as follows:

*Figure 2 (A) The UGS-BVOC emissions of each species (upper panel) and relative difference (Ghr - Gdef) from various land use cover (lower panel), (B) the proportion of the BVOC emissions from urban and nature areas (upper panel) and the relative proportion difference (Ghr - Gdef) from various land use cover (lower panel), (C) the relative difference of solar radiation (C), and (D) surface temperature in each region driven via various land use cover datasets. All values in these figures are during September 2017.*

**Comment 12**: l. 365: Figures should be numbered in the order as they are cited in the text. As you refer several times in this paragraph to Figure 5, you should move this figure before Figure 3.

**Reply:** Thank you for your careful review. We have adjusted the figure numbering and placement to ensure that figures are cited in the correct order, moving Figure 5 before Figure 3 as recommended.

**Comment 13**: l. 381: 'delves' is an overused verb since the advent of large language models. Simply replace by 'shows'.

**Reply:** Thank you for your careful review. I have changed the `delves` to `shows`.

*Additionally, Figure 4C shows into the disparities in the UGS-BVOC emissions attributed to different land use cover datasets.*

**Comment 14**: l. 439: calling an increase of ~3% 'great' seems an exaggeration. I suggest removing 'greatly' here.

**Reply:** Thanks for this valuable suggestion. We have removed 'greatly' here.

*The analysis reveals that the UGS-BVOC emissions alone (Figure 6A) primarily affect the city center region, increasing MDA8 $O_3$ concentrations by 1.0-1.4 ppb (+2.3-3.2%), ...*

**Comment 15**: l. 487/8: This sentence is quite convoluted. Please simplify.

**Reply:** Thanks for the valuable suggestion. We have rephrased this sentence as follows:

*Figure 8 presents the assessment of O3 episode simulations. The Gdef_N case initially underestimated O₃ concentrations, leading to an evaluation of improvements using three cases: UGS-LUCC (Ghr_N), UGS-BVOC emissions (Gdef_Y), and their combined effects (Ghr_Y).*

**Comment 15**: - I suggest merging 'Sections 4 and 5' into a a coherent concluding section. Please make sure to include all elements as requested in our journal guidelines

https://www.atmospheric-chemistry-and-physics.net/policies/guidelines_for_authors.html

**Reply:** Thanks for the valuable suggestion. We have merged 'Sections 4 and 5' into a coherent concluding section.

**In Section 4: Conclusion:**

*However, some uncertainties and limitations remain in this study. First, the 10-m resolution land use and land cover data still cannot fully capture the spatial pattern of UGS in Guangzhou. As shown in Figure S2, although UGS in Guangzhou is primarily composed of EBTs, most of these EBTs are distributed along urban edges. This may result from distortions in the definition of urban extent, such as misclassifying mixed urban-vegetation grids as urban grids, caused by the coarse resolution of the 1-km land use and land cover data. The fuzzy definition of urban boundaries could lead to non-UGS areas being misclassified as UGS, potentially resulting in an overestimation of UGS-BVOC emissions. Second, due to resolution limitations, only larger patches of grassland, cropland, and woodland are recognized as UGS, while smaller UGS vegetation, such as street trees, often goes undetected at a 10-m resolution. This omission can lead to an underestimation of the UGS-BVOC emissions. Third, the 10-m and 1-km resolution land use and land cover data, along with the growth forms and ecotype data, use simplified categorizations for grids, which cannot fully capture the diversity of vegetation species within UGS. Since different vegetation species have varying emission factors, this simplification introduces some errors. Similarly, the oversimplified classification of land grids limits this study's ability to provide specific planning strategies for UGS at the species level. Nevertheless, it can highlight the importance of considering UGS-BVOC and UGS-LUCC in air pollution prevention and control policies. Finally, Guangzhou, the study area, is a highly urbanized Chinese metropolis with a VOC-limited region (Gong et al., 2018; Liu et al., 2018; Zhou Kai et al., 2011). As a result, even a relatively small amount of VOC emissions, such as those from UGS-BVOC, can significantly impact ozone concentrations. Therefore, policymakers in Guangzhou should prioritize addressing the role of UGS-BVOC emissions in air pollution prevention and control. In other cities, particularly those with advanced urban development, high $NO_x$ emissions—often resulting from factors like high motor vehicle ownership—can lead to VOC-limited conditions. In such areas, it is equally important to emphasize the role of UGS-BVOC emissions in ozone pollution. In contrast, cities with lower $NO_x$ emissions identified as $NO_x$-limited regions may experience minimal impact from UGS-BVOC emissions on ozone concentrations.*